# STEF/TIAM2-mediated Rac1 activity at the nuclear envelope regulates the perinuclear actin cap

Anna Woroniuk[1], Andrew Porter[1], Gavin White[1], Daniel T. Newman[2], Zoi Diamantopoulou[1], Thomas Waring[2], Claire Rooney[1], Douglas Strathdee[3], Daniel J. Marston[4], Klaus M. Hahn[4], Owen J. Sansom [3,5], Tobias Zech [2] & Angeliki Malliri[1]

The perinuclear actin cap is an important cytoskeletal structure that regulates nuclear morphology and re-orientation during front-rear polarisation. The mechanisms regulating the actin cap are currently poorly understood. Here, we demonstrate that STEF/TIAM2, a Rac1 selective guanine nucleotide exchange factor, localises at the nuclear envelope, co-localising with the key perinuclear proteins Nesprin-2G and Non-muscle myosin IIB (NMMIIB), where it regulates perinuclear Rac1 activity. We show that STEF depletion reduces apical perinuclear actin cables (a phenotype rescued by targeting active Rac1 to the nuclear envelope), increases nuclear height and impairs nuclear re-orientation. STEF down-regulation also reduces perinuclear pMLC and decreases myosin-generated tension at the nuclear envelope, suggesting that STEF-mediated Rac1 activity regulates NMMIIB activity to promote stabilisation of the perinuclear actin cap. Finally, STEF depletion decreases nuclear stiffness and reduces expression of TAZ-regulated genes, indicating an alteration in mechanosensing pathways as a consequence of disruption of the actin cap.

[1] Cell Signalling Group, Cancer Research UK Manchester Institute, The University of Manchester, Alderley Park SK10 4TG, UK. [2] Cellular and Molecular Physiology, Institute of Translational Medicine, University of Liverpool, Liverpool L69 3BX, UK. [3] Cancer Research UK Beatson Institute, Garscube Estate, Switchback Road, Glasgow G61 1BD, UK. [4] Department of Pharmacology, University of North Carolina, Chapel Hill, NC 27599-7365, USA. [5] Institute of Cancer Sciences, University of Glasgow, Glasgow G61 1BD, UK. Correspondence and requests for materials should be addressed to A.M. (email: Angeliki.Malliri@cruk.manchester.ac.uk)

Control of nuclear shape is important for many dynamic cellular processes and aberrant nuclear morphology is a feature of a number of cancer sub-types[1–4]. The actin cap, a contractile structure composed of thick, aligned actomyosin filaments that interact with the apical surface of the interphase nucleus via the Linker of the Nucleus and Cytoskeleton (LINC) complex, is a key regulator of nuclear morphology[1,2,5]. Filaments of the actin cap can be distinguished from other stress fibres, such as basal stress fibres or the transverse arcs that constitute trans-membrane actin-associated nuclear (TAN) lines, both through their parallel orientation with the cellular axis and their termination at specific actin cap associated focal adhesions (ACAFA)[6]. The functional roles of this recently discovered cytoskeletal organelle are expanding. The contractile structure of the cap functions to constrain the nucleus, coupling nuclear and cellular geometry, which is particularly important in migrating cells[2]. In addition, the perinuclear actin cap co-operates with the inter-mediate filament network to oppose dynein-microtubule driven rotation of the interphase nucleus[1,7–9] and an intact actin cap is required for efficient re-orientation of the nucleus in polarising fibroblasts[10]. The tension force required to constrain and anchor the nucleus is dependent on actomyosin contractility generated by the actin motor Non-muscle myosin IIB (NMMIIB); depletion of NMMIIB causes nuclear expansion and an over-rotation phenotype[11]. There is mounting evidence that the actin cap works to bridge the extracellular environment and the nucleus via its connection with ACAFA and the LINC complex, providing a direct, rapid pathway for mechanotransduction[5,6,12]. Stress fibres of the actin cap exhibit greater sensitivity to changes in substrate compliance or shear stress than basal stress fibres, and formation of the actin cap has been shown to regulate chromatin arrange-ment in the nucleus, supplying further evidence that the actin cap is essential for providing mechanical continuity for rapid signal transmission to regulate gene expression and elicit cellular responses[6,13,14].

The molecular basis of actin cap formation and stability is poorly understood. Actin regulatory proteins are known to be involved, including the refilin/filamin proteins. Refilin B recruits filamin A to pre-existing perinuclear actin cables and converts its activity from actin branching to actin bundling, promoting the formation of the actin cap[15,16]. The Rho GTPase family of pro-teins are strongly associated with the regulation of cytoskeletal dynamics[17,18], and there is a proposed role for RhoA in the generation of perinuclear actin stress fibres and ROCK-mediated regulation of actomyosin contractility[2,19]. Lipid modifications at the C-terminal polybasic region of Rho GTPases enable their association with membranes, where they can be regulated by guanine nucleotide exchange factors (GEFs)/GTPase activating proteins (GAPs) and interact with effectors[20–22]. Rac1 has been shown to be localised at the nuclear membrane[20], and utilisation of a FLAIR (FLuorescence Activation Indicator for Rho proteins) Rac1 biosensor revealed an activation of juxtanuclear Rac1 in migrating fibroblasts[23]. However, very little is known about the regulation or the functional role of localised Rac1 activity at the nuclear envelope.

Here, we identify that the Rac1 selective GEF Sif and TIAM1-like Exchange Factor STEF, also known as TIAM2, localises to the nuclear envelope, co-localises with key perinuclear proteins and regulates the activity of perinuclear Rac1. We show that down-regulation of STEF-mediated Rac1 activity at the nuclear envelope results in a disruption of the perinuclear actin cap, increased nuclear height and an impairment of nuclear re-orientation during front-rear polarisation. Moreover, we observe a decrease in pMLC and myosin-generated tension at the nuclear envelope in STEF-depleted cells, indicating that localised STEF-mediated Rac1 activity might regulate NMMIIB activity to promote

stabilisation of the actin cap. Consistent with these data, STEF depletion results in a decrease in nuclear stiffness and reduced expression of TAZ-regulated genes, indicating that mechan-osensing pathways are altered.

## Results

**The Rac activator STEF localises to the nuclear envelope.** We observed that endogenous STEF localises to the nuclear envelope of U2OS cells, with fluorescence intensity profiles displaying two clear peaks of STEF signal intensity correlating with the nuclear periphery, as verified by DAPI staining (Fig. 1a, b). Two inde-pendent STEF-depleted CRISPR clones (CRISPR#1 and CRISPR#2) were generated in the U2OS cell line and validated by SURVEYOR assay, which revealed the targeted disruption of one STEF allele (Supplementary Fig. 1a, b). The STEF-depleted CRISPR clones exhibited a substantial reduction in fluorescence intensity for STEF, with a loss of the nuclear ring localisation, indicating that the fluorescence signal is specific for STEF (Fig. 1a, Supplementary Fig. 1c, d). The perinuclear localisation of endo-genous STEF was observed in multiple cell types including mouse embryonic fibroblasts (MEFs), COS-7 cells and lung adeno-carcinoma A549 cells (Fig. 1c). Furthermore, exogenous full-length STEF (FL-STEF) clearly exhibited a perinuclear localisa-tion (Fig. 1d). Interestingly, we also observed what appears to be intra-nuclear STEF, including puncta, which are reduced in the CRISPR clones (Fig. 1a); however, this study focuses on the role of the perinuclear pool of STEF.

To determine the structural domain of STEF responsible for this interesting perinuclear localisation, we analysed the localisa-tion of a range of deletion mutants of STEF in U2OS cells and compared their localisation to exogenous FL-STEF (Fig. 1d–f). Deletion of the STEF N-terminus (ΔN-STEF) did not affect localisation to the nuclear membrane (Fig. 1d). In addition, deletion of the CC-EX, PDZ, DH or the N-terminal PH domain (PHn) of STEF did not affect its ability to localise to the nuclear membrane (Fig. 1e, f); however, deletion of its C-terminal PH domain (PHc) abolished its perinuclear localisation (Fig. 1e, f). Additionally, we observed that a truncated form of STEF containing only the DH-PHc domains does localise at the nuclear envelope (Fig. 1e, f). Taken together these results indicate the importance of the C-terminal PH domain in mediating the localisation of STEF at the nuclear envelope. Fractionation of total-cell lysates, as performed in ref [24], confirmed a substantial pool of perinuclear-enriched STEF in both U2OS and MEFs (Fig. 1g and Supplementary Fig. 1e). To our knowledge, this is the first identification of a GEF for Rac1 at the nuclear envelope, and interestingly this localisation is not shared by the close homologue of STEF, TIAM1 (Supplementary Fig. 1f, g). To investigate the localisation of STEF in relation to the structure of the nuclear envelope, U2OS cells were treated with digitonin to selectively permeabilise the cholesterol-rich plasma membrane while leaving the nuclear membrane intact[25]. We then probed with antibodies for STEF and Lamin A/C (which localises on the interior of the nuclear envelope) (Fig. 1h). We observed substantial staining for STEF in digitonin-permeabilised cells, with no staining for the intranuclear control Lamin A/C, indicating the presence of a pool of STEF at the cytoplasmic face of the outer nuclear membrane (Fig. 1h).

**STEF indirectly interacts with Nesprin-2G and NMMIIB.** To corroborate the perinuclear localisation of STEF, we investigated the co-localisation of STEF with key proteins of the perinuclear region. Nesprin-2G is a giant structural scaffolding protein anchored in the outer nuclear envelope, with a crucial role in coupling the nucleus to the overlying actin network for the

regulation of nuclear positioning, re-orientation and mechanosensing pathways[5,12,26–28]. Co-staining of U2OS cells showed a high degree of co-localisation between STEF and Nesprin-2G, with both proteins exhibiting a characteristic nuclear ring localisation and similar fluorescence intensity profiles (Fig. 2a, b). To further characterise this co-localisation, we performed the

Duolink® proximity ligation assay (Fig. 2c, d); we saw puncta when using antibodies against both STEF and Nesprin-2G, indicating that these two proteins are closely localised (typically less than 40 nm between the two antibodies is required to generate a signal). We assessed the nature of the interaction between STEF and Nesprin-2G in co-immunoprecipitation assays. We

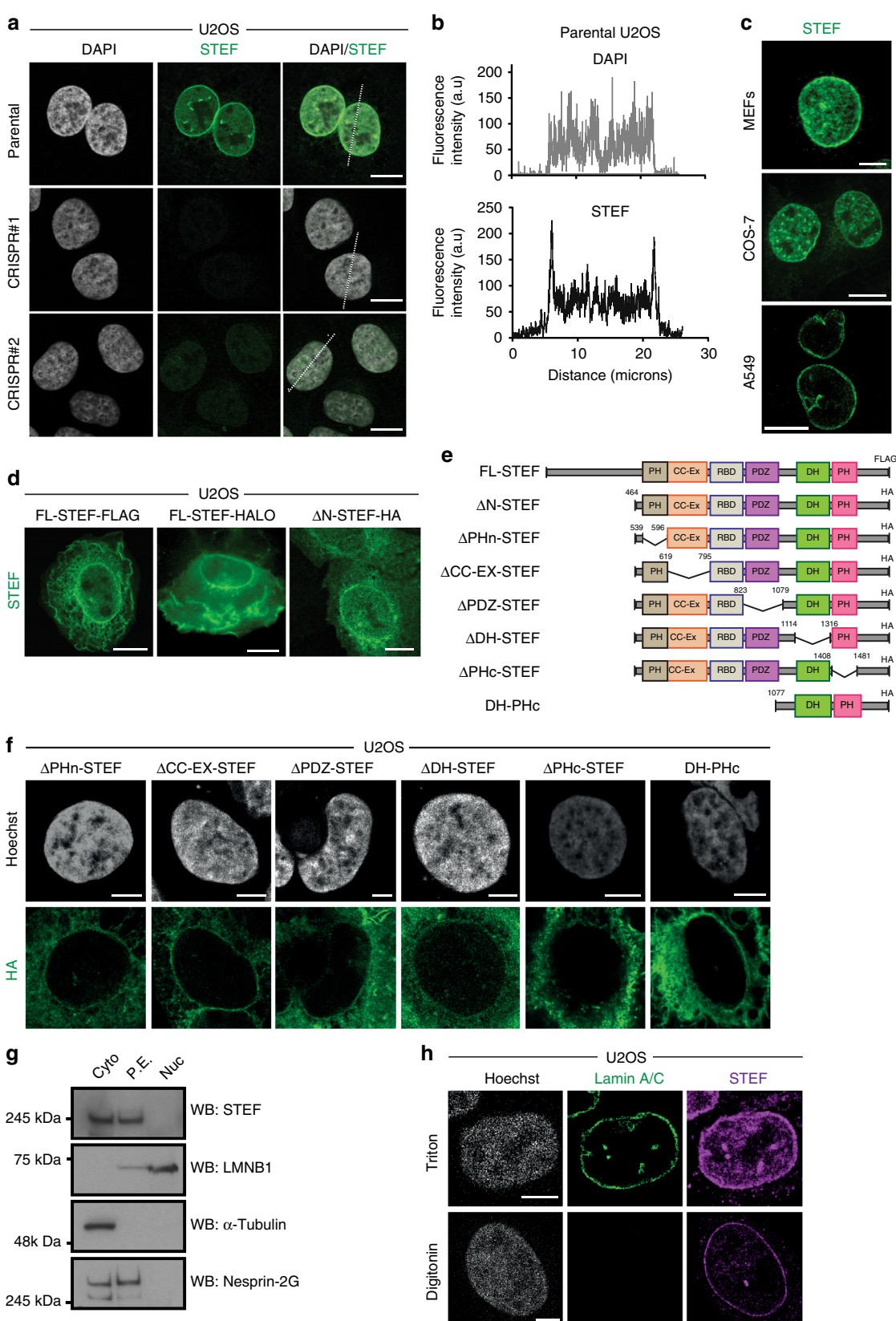

confirmed the specificity of our antibody by knocking down endogenous Nesprin-2G with two distinct siRNAs (Supplementary Fig. 2a) and by comparing the size of the Nesprin-2G band with that of another large protein, Huwe1 (482 kDa) (Supplementary Fig. 2b). Pull-down of endogenous STEF from U2OS cell lysates efficiently co-precipitated endogenous Nesprin-2G (Supplementary Fig. 2c). Due to the large size of Nesprin-2G (796 kDa), previous experiments to probe its function have often utilised the truncated mini-Nesprin-2G (mN2G) construct, which is composed of the minimal functional domains of the giant-form: the N-terminal Calponin homology domains and first two spectrin repeats fused to the last two spectrin repeats and the C-terminal transmembrane and KASH domains[27,29] (Supplementary Fig. 2d). Interestingly, although mN2G localised in the nuclear envelope (Supplementary Fig. 2e), we did not observe an interaction between STEF and mN2G through either conventional co-immunoprecipitation or GFP-trap techniques, indicating the importance of the spectrin repeat domains in mediating the interaction with STEF. Moreover, endogenous immunoprecipitated Nesprin-2G did not pull-down recombinant STEF, indicating that the interaction is indirect (Supplementary Fig. 2f), and that the proteins may be part of a larger complex.

NMMIIB is another key component of the actin cap, where it functions as a crucial regulator of actomyosin contractility[30]. We observed that it localises to perinuclear filaments in U2OS cells (Fig. 2e), and that there are points of co-localisation between STEF at the nuclear membrane and NMMIIB filamentous staining (indicated by the arrow heads in the magnified insert). We further showed that pull-down of endogenous STEF from U2OS cell lysates efficiently co-precipitated endogenous NMMIIB (Fig. 2f). We examined if the interaction between STEF and NMMIIB occurs directly using an in vitro assay; however, while STEF was efficiently pulled down by the positive control PAR3[31], recombinant full-length STEF was not pulled-down by endogenous immunoprecipitated NMMIIB, indicating that STEF and NMMIIB interact indirectly (Supplementary Fig. 2g). Overall these results support the localisation of STEF at the outer nuclear envelope in close proximity to key perinuclear protein complexes containing NMMIIB and Nesprin-2G.

**STEF is a key regulator of perinuclear actin dynamics**. The identification of a GEF for Rac1 that co-localises with key perinuclear actin-binding and actin-regulating proteins led us to hypothesise that STEF regulates actomyosin contractility at the nuclear envelope. Rac1 is a critical regulator of actin dynamics through both Arp2/3 complex and formin-mediated actin polymerisation pathways[32–34], and there is also increasing evidence of a role for Rac1 in the regulation of myosin II activity[35–37]. Utilising the Cre/LoxP system we generated inducible *Stef* knockout mice and isolated MEFs. These were further modified with doxycycline-inducible expression of STEF (pRETROXT-Pur-

STEF-HALO) for subsequent rescue experiments (Fig. 3a, Supplementary Fig. 3a–c). Infection of MEFs with adenoviral Cre resulted in a substantial reduction in STEF expression levels (STEF KO) relative to cells infected with control virus (control), which could be rescued through re-expression of wild-type (WT) STEF via addition of doxycycline (STEF KO + WT STEF) (Fig. 3a).

To investigate the effect of STEF depletion on perinuclear actin, sparsely plated MEFs (treated as in Fig. 3a) were stained with Phalloidin and the number of apical actin cables overlying the nuclear surface was quantified (Fig. 3b, c). In control MEFs, many cells exhibited a well-formed actin cap, with numerous linear cables evident in the z-planes above the apical nuclear surface (Fig. 3b, c). However, STEF-depleted MEFs (STEF KO) exhibited a substantial reduction in the number of apical actin cables overlying the nucleus (Fig. 3b, c). This reduction could be rescued through re-expression of WT STEF (STEF KO + WT STEF) (Fig. 3b, c). Depletion of STEF specifically disrupted the perinuclear actin cap, as we observed no substantial change in the number or organisation of basal stress fibres (Supplementary Fig. 3d, e). The impairment in perinuclear actin was also observed in STEF-depleted U2OS cells (Supplementary Fig. 3f, g). To determine the dependency of this phenotype on the GEF activity of STEF, as opposed to a purely scaffolding function of STEF, conditional STEF KO MEFs were generated with inducible expression of a GEF-mutant STEF (pRETROXT-Pur-STEF-HALO-DH*), (Fig. 3d) which cannot activate Rac1[38]. Expression of the GEF-mutant construct did not rescue the perinuclear actin cable phenotype in STEF-depleted cells (STEF KO + STEF DH*) (Fig. 3e, f). These results demonstrate that the GEF activity of STEF is required for the formation and/or stabilisation of the perinuclear actin cables of the actin cap.

**Localised Rac1 activity regulates the actin cap**. The failure of the GEF-dead form of STEF to rescue the disruption of the actin cap in STEF-depleted cells suggested that STEF-mediated Rac1 activation is essential for the perinuclear actin cap. Previously, RhoA activity has been implicated in the regulation of the perinuclear actin cap[2,19]. We therefore tested the effect of STEF depletion on the activity of RhoA as well as Rac1 and Cdc42 GTPases using an ELISA assay. Depletion of STEF resulted in a substantial decrease in Rac1 activity, with no effect on the activity of RhoA and Cdc42 (Supplementary Fig. 4a), consistent with previously published work showing that STEF is a Rac1 specific GEF[39] and also work demonstrating that down-regulation of STEF results in a global decrease in Rac1 activity[38]. To determine if STEF regulates perinuclear Rac1 activity we used a Rac1 Fluorescence Resonance Energy Transfer (FRET) biosensor. A Raichu-Rac1 FRET probe was expressed in parental U2OS cells and two STEF-depleted CRISPR clones (CRISPR#1 and CRISPR#2), where it localised throughout the cytoplasm and around the nuclear periphery

**Fig. 1** The Rac activator STEF localises at the nuclear envelope. **a** Representative high-resolution confocal immunofluorescence images of parental U2OS cells and two U2OS CRISPR clones stained for DNA (DAPI) and STEF. **b** Fluorescence intensity profiles of DAPI and STEF signals across the nuclei of U2OS parental cells. Position of line scan indicated by the dashed white line in **a**. **c** Immunofluorescence images of endogenous STEF expression in mouse embryonic fibroblasts (MEFs) (top), primate fibroblast-like COS-7 cells (middle) and human lung adenocarcinoma A549 cells (bottom). **d** Immunofluorescence images of STEF expression in U2OS cells transfected with pcDNA3-FL-STEF-FLAG (left), pRETROXT-WT STEF-HALO (middle) or pcDNA3-ΔNSTEF-HA expressing a truncated STEF mutant (right). **e** Schematic representation of the domain structure of full-length STEF and a range of deletion mutant constructs of STEF. **f** Representative confocal images of U2OS cells transfected with a range of deletion mutant constructs of STEF (all in the pcDNA3 vector, with a C-terminal HA tag), stained for DNA (Hoechst) and HA. **g** U2OS cells were fractionated into cytoplasmic (Cyto), perinuclear-enriched (P.E.) and core-nuclear (Nuc) extracts using a successive lysis protocol. Lysates from each fraction were prepared containing equal total protein, and probed for expression levels of STEF alongside Tubulin (as a marker of the cytoplasmic fraction), Lamin B1 (LMNB1, as a marker of the core-nuclear fraction) and Nesprin-2G (as a marker of the perinuclear-enriched fraction). **h** Representative confocal images of parental U2OS cells permeabilised either with triton (top panels) or digitonin (bottom panels) and stained for STEF, DNA (Hoechst) and Lamin A/C. Scale bars throughout = 10 μm, except in (h) = 5 μm

(Fig. 4a). Measurement of the FRET ratio in the perinuclear region showed a substantial decrease in perinuclear Rac1 activation in the STEF-depleted CRISPR clones compared to controls (Fig. 4b), confirming that STEF regulates the activity of Rac1 adjacent to the nuclear envelope.

To further investigate the role of perinuclear Rac1 activity in the formation and/or maintenance of the actin cap, we generated

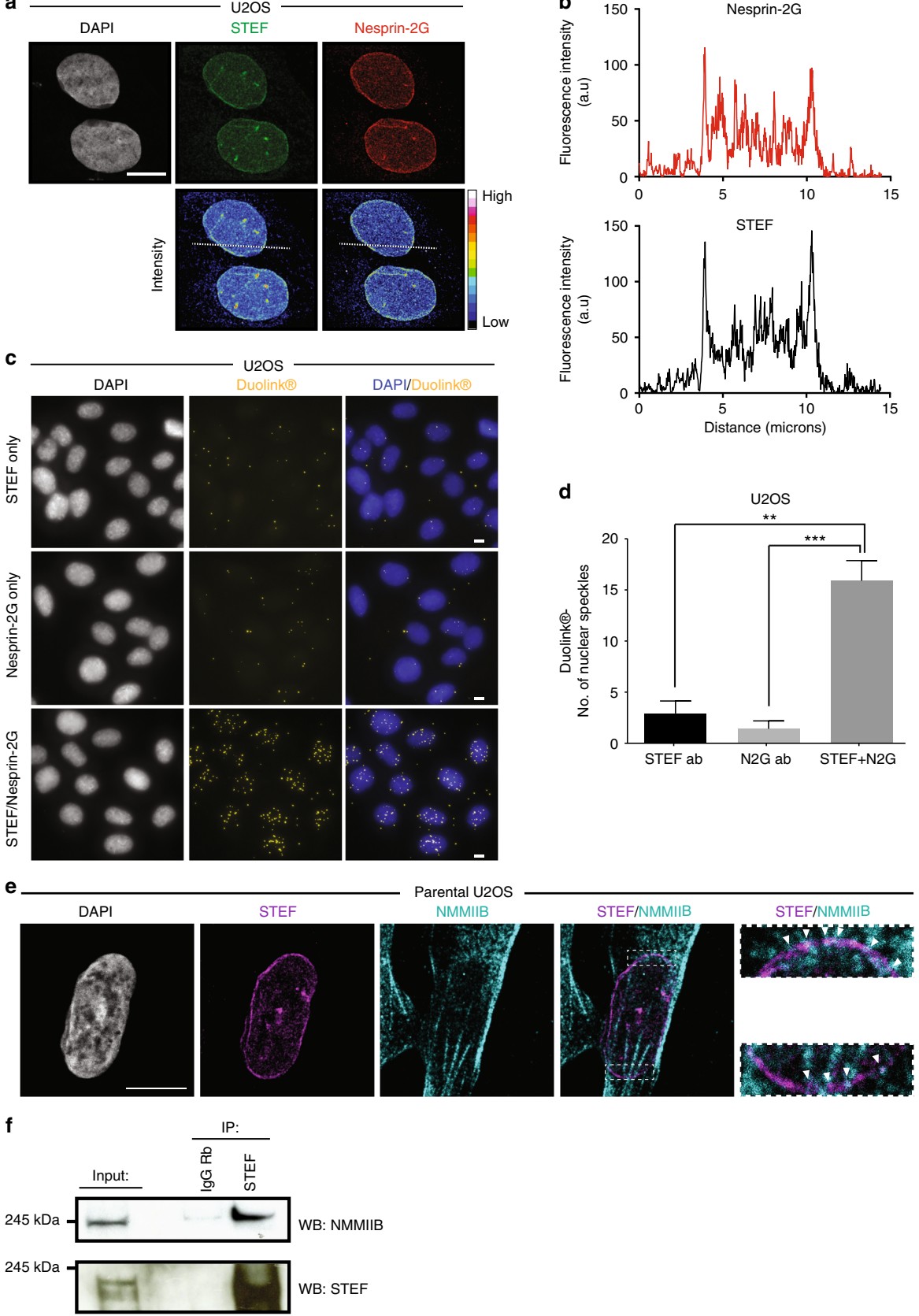

constructs to target Rac1 to the nuclear envelope. The KASHext domain, based on the KASH domain of Nesprin-2α with an additional 13 C-terminal amino acids, retains the perinuclear targeting ability of the WT KASH domain without disrupting the Nesprin-SUN2 interaction[40]. We fused the KASHext domain onto the C-terminus of eGFP-WT-Rac1, eGFP-V12-Rac1 (constitutively active Rac1) and eGFP-N17-Rac1 (dominant negative Rac1) (Fig. 4c) and verified that all constructs localised to the nuclear envelope (Supplementary Fig. 4b). We also verified that WT and V12 constructs expressed to the same level as measured by western blotting and fluorescence intensity of individual cells (Supplementary Fig. 4c and d). Targeting constitutively active-Rac1 to the nuclear envelope in STEF KO MEFs was sufficient to rescue the disruption of the perinuclear actin cap caused by loss of STEF (Fig. 4d, e) with no change observed in basal actin fibres (Supplementary Fig. 4e). However, expression of WT-Rac1 at the nuclear envelope was unable to rescue the phenotype of STEF KO MEFs, highlighting the requirement of STEF to activate Rac1 at the nuclear periphery and suggesting that no other Rac1 GEF is able to compensate for the loss of STEF (Fig. 4d, e). Additionally, we expressed the dominant-negative form of Rac1 at the nuclear envelope in WT MEFs. N17-Rac1 commonly exists in a nucleotide-free state, binding strongly to GEFs and preventing them from activating endogenous Rac1. Localised expression of N17-Rac1 at the nuclear envelope caused a substantial disruption of the actin cap (Fig. 4f, g). These experiments confirm that STEF regulates the perinuclear actin cap via Rac1 activation.

**STEF regulates nuclear re-orientation and height**. The contractile actin cap has been shown to constrain nuclear morphology and be required for re-orientation of the nucleus during the establishment of front-rear polarity[2,10]. Fibroblast nuclei display a restricted rotation in the $x/y$ plane during the establishment of front-rear polarity, such that the longitudinal axis of the nucleus is re-oriented with the cellular axis of future migration[26]. Disruption of this nuclear re-orientation process has been shown to impair directionality of migration, indicating it to be a critical priming step for periods of cellular movement[26,41]. Given our data above showing a reduction in perinuclear actin cables following STEF depletion, we further examined the nuclear characteristics of STEF-depleted MEFs. Control, STEF KO, or STEF KO + WT STEF MEFs were plated onto collagen-coated crossbow shaped micropatterns, which provide the adhesive patterning required to recapitulate the classic front-rear polarised phenotype[42]. Following plating, cells were allowed to adhere and polarise, before fixation and staining for the nucleus (DRAQ5) and actin cytoskeleton (Phalloidin) (Fig. 5a). Representative images of control MEFs demonstrate characteristic front-rear polarised morphology (Supplementary Fig. 5a). We quantified the nuclear angle relative to the cellular axis in polarised MEFs (Fig. 5b). There was a noticeable failure of nuclear orientation to align with the cellular axis in STEF KO MEFs, which could be rescued through re-expression of WT STEF (Fig. 5c). This defect

in nuclear re-orientation was also observed in STEF-depleted U2OS CRISPR clones undergoing polarised migration into an in vitro wound during a scratch assay (Supplementary Fig. 5b–d). Moreover, tracking individual cells revealed a defect in the directionality of migration of U2OS cells with depleted STEF (Supplementary Fig. 5e–h) consistent with the inability of these cells to re-orient their nuclei and the reduced migration of STEF knockdown cells we have previously reported [38].

The physical dimensions of the nucleus reflect the functionality of the perinuclear actin cap, as disruption of the cap results in an increase in nuclear height[11]. Therefore, we measured the height of the nucleus in MEFs on micro-patterned plates and found that STEF depletion induced a substantial increase in the height of the nucleus, which was rescued by WT STEF (Fig. 5d, e). Taken together, these results indicate that the disrupted actin cap in STEF-depleted cells translates into impaired constraint of the nucleus, which alters nuclear morphology and nuclear re-orientation during the establishment of front-rear polarity.

**STEF regulates nuclear stiffness and perinuclear tension**. Nuclear stiffness is regulated by the actin cap as well as by the underlying lamin network and chromatin[2,43,44]. To assess whether STEF depletion affects nuclear rigidity, we used atomic force microscopy (AFM) to measure the elastic modulus of the nucleus in control and STEF KO MEFs and identified a substantial reduction in the Young's Modulus of STEF KO MEFs relative to the control MEFs (Fig. 6a). This indicates that disruption of the actin cap in STEF KO MEFs results in a softer nucleus. To confirm that this decrease in nuclear stiffness is not due to a perturbation in the expression of the intranuclear lamin proteins, we analysed the expression of Lamin A/C and B1 in control and STEF-depleted MEFs. Western blotting showed no change in the expression of Lamin A/C (LMNA/C) and B1 (LMNB1) upon depletion of STEF (Supplementary Fig. 6a, b) and immuno-fluorescence imaging revealed no change in the distribution of Lamin A/C following STEF depletion (Supplementary Fig. 6c).

To investigate further the role of STEF-mediated Rac1 activation in perinuclear actin cap formation and/or maintenance, we examined actomyosin contractility at the nuclear envelope. Tension sensors provide a useful tool for determining whether a specific protein is subject to cytoskeletal-generated force. The development of a mini-Nesprin-2G tension sensor (mN2G-TS) revealed that Nesprin-2G is subject to actomyosin-generated tension, transducing force from the perinuclear cytoskeleton to the nuclear surface[45]. We utilised mN2G-TS as a reporter of actomyosin-generated tension at the nuclear envelope to determine whether STEF depletion and the consequent impairment of the actin cap affects perinuclear tension. The mN2G-TS is a FRET based reporter with a Teal Fluorescent Protein (TFP)-Venus donor/acceptor pair (Fig. 6b)[45,46]. Binding of overlying actin cables to the N-terminal actin-binding domain of the sensor induces tension, resulting in a conformational change of the protein that leads to a reduction in the FRET index (Fig. 6b)[45]. In

**Fig. 2** STEF co-localises and interacts with Nesprin-2G and NMMIIB. **a** High-resolution confocal immunofluorescence images of parental U2OS cells co-stained for DNA (DAPI), STEF and Nesprin-2G, with an intensity look-up table (16-colours, ImageJ). **b** Fluorescence intensity profile of STEF (black) and Nesprin-2G (red) signals across the nuclei of U2OS parental cells. Position of line scan indicated by the dashed white line in **a**. **c** The Duolink® PLA assay was conducted in parental U2OS cells with the Duolink® II red kit, and cells were imaged in the red fluorescence channel. Immunofluorescence images of the fluorescent signal obtained for single primary antibody controls (STEF only, Nesprin-2G only) and interaction between STEF and Nesprin-2G (STEF/ Nesprin-2G). **d** Quantification of Duolink® signal from primary antibody controls and STEF-Nesprin-2G antibodies ( >100 cells per condition per replicate). Values represent the mean of three independent experiments. Statistical significance was verified using a one-way ANOVA, using Tukey's multiple comparison test to compare the means of each sample. ** $p < 0.01$, *** $p < 0.001$. Error bars represent S.E.M. **e** Immunofluorescence images taken on the high-resolution confocal microscope of parental U2OS cells stained for DNA (DAPI), STEF and NMMIIB. White arrowheads indicate co-localisation between STEF and NNMIIB. **f** Co-immunoprecipitation (IP) of endogenous NMMIIB with endogenous STEF from parental U2OS cell lysates, compared to the IgG control. Representative western blot shown from four independent immunoprecipitation experiments. Scale bars = 10 μm throughout

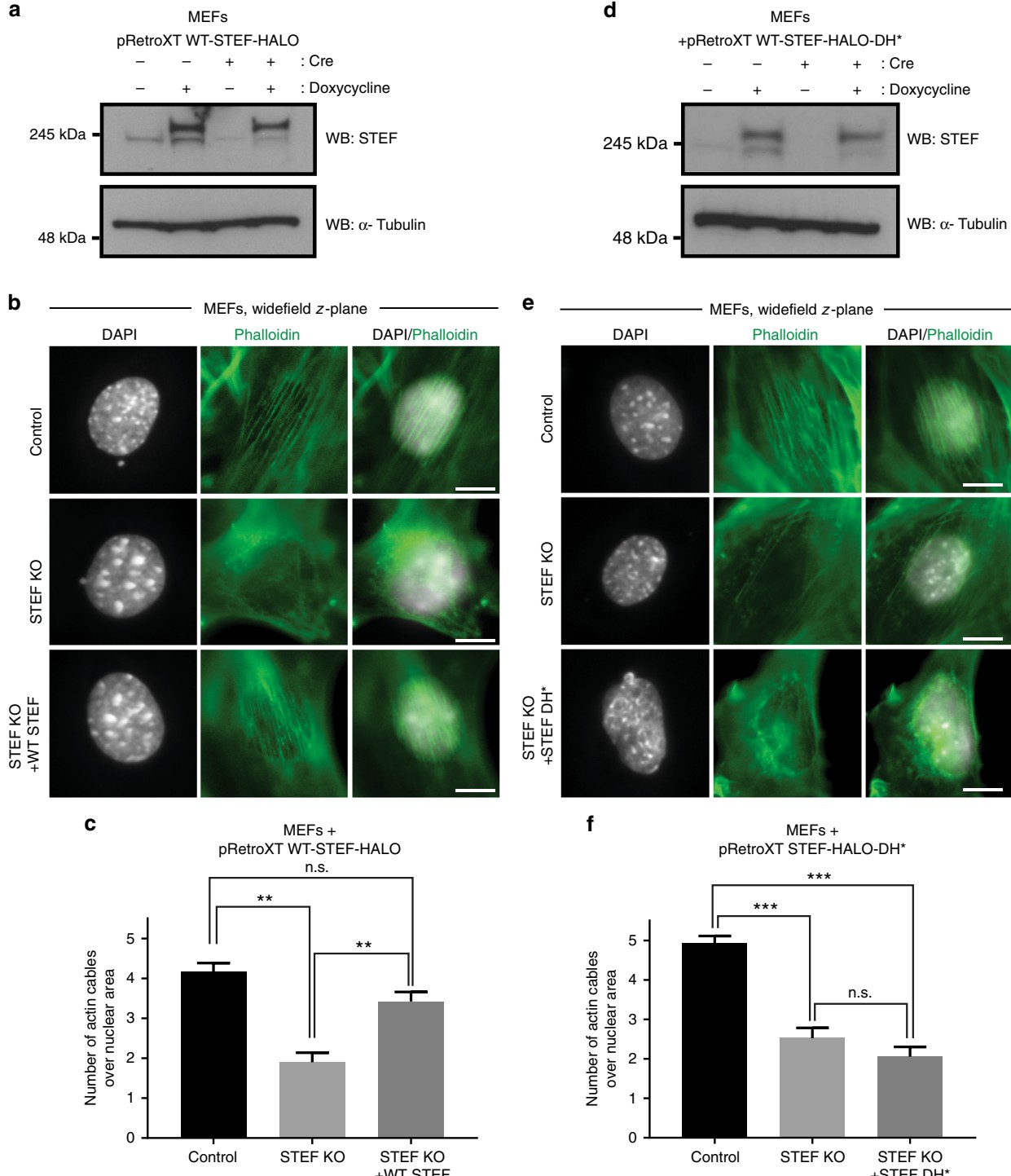

**Fig. 3** STEF depletion disrupts the apical perinuclear actin cables. **a**, **d** Conditional STEF KO MEFs with inducible expression of either pRETROXT-Pur-STEF-HALO (**a**) or pRETROXT-Pur-STEF-DH*-HALO (**d**) were generated and pre-incubated in the presence or absence of doxycycline for 24 h, prior to infection with either adenoviral-Cre ( + Cre) to deplete STEF or an empty adenoviral control (-Cre). Lysates were harvested after 72 h and STEF expression was determined by western blotting. α-Tubulin was used as a loading control. **b**, **e** Representative widefield immunofluorescence images of sparsely plated MEFs treated as in **a** and **d**, stained for DNA (DAPI) and F-actin (Phalloidin). Scale bar = 10 μm. **c**, **f** Quantification of apical actin cable number over nuclear area from cells as in **b** and **e**. Values represent the mean of three independent experiments ± S.E.M ( >25 cells per condition, per replicate). Statistical significance was verified using a one-way ANOVA, using Tukey's multiple comparison test to compare the means of each sample. ** $p < 0.01$, *** $p < 0.001$, n.s. = not significant

control and STEF-depleted MEFs, the mN2G-TS construct correctly localised to the nuclear envelope, similar to endogenous Nesprin-2G (Fig. 6c). As a positive control, we treated control MEFs with a combination of a Myosin light-chain kinase inhibitor (ML7) and a ROCK inhibitor (Y-27632) in order to ablate myosin-generated tension[45], which substantially increased the FRET index in the control MEFs (Fig. 6d). In STEF-depleted MEFs, we observed a substantial increase in the FRET index,

indicating a reduction in tension on the construct (Fig. 6c, d) compared with cells expressing WT levels of STEF.

To further investigate the effect of depleting STEF on actomyosin contractility, we analysed the number of pMLC-

positive cables overlying the nucleus of control and STEF-depleted MEFs as an indicator of localised NMMIIB activity at the nuclear envelope (Fig. 6e). We saw a substantial reduction in the number of pMLC-positive cables in STEF-depleted MEFs in

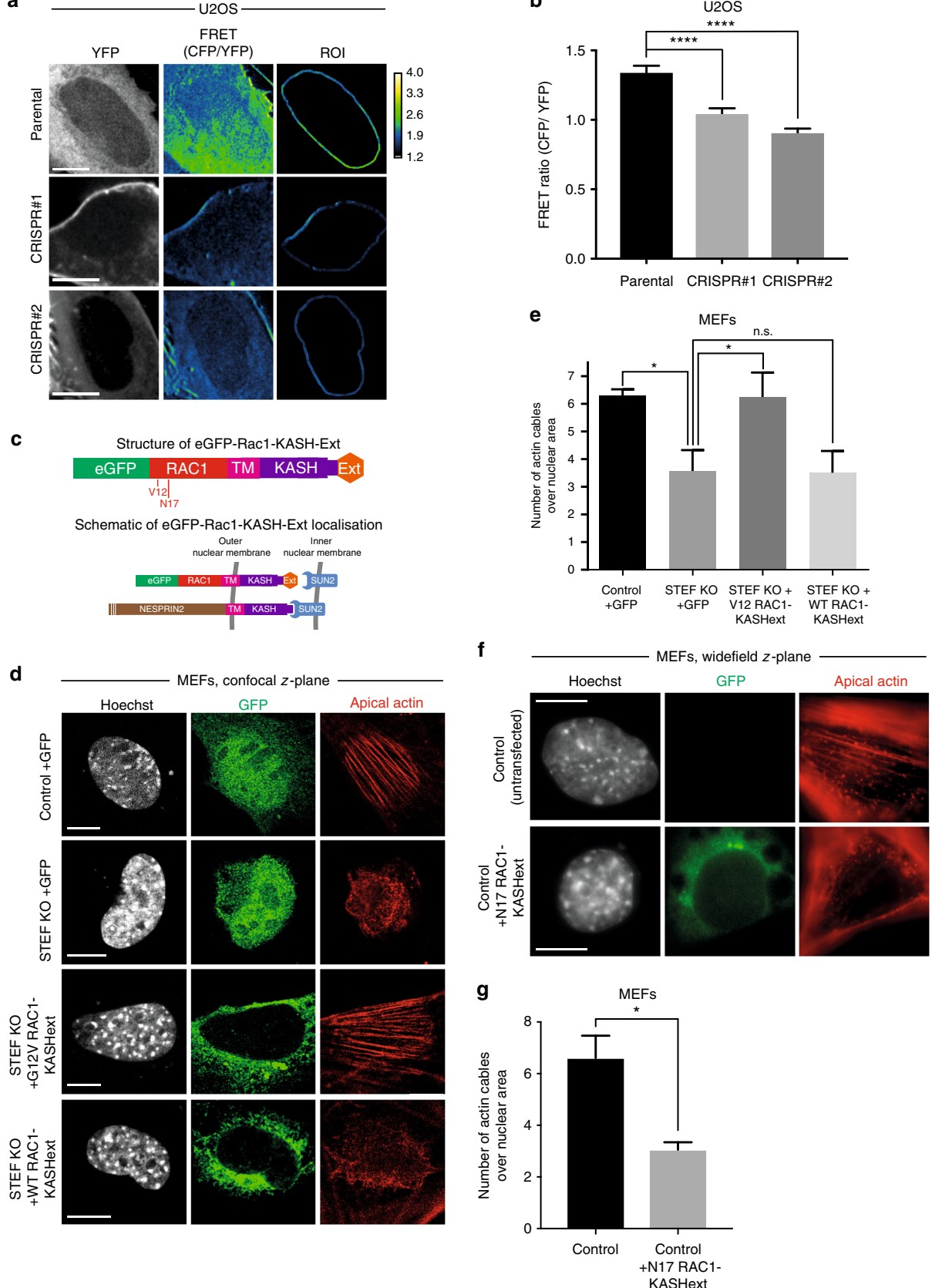

comparison with control MEFs, indicating substantially lower levels of active NMMIIB at the nuclear envelope (Fig. 6e, f). These results indicate that the myosin-dependent tension on the LINC complex is considerably reduced by depletion of STEF, consistent with the impaired actin cap formation of these cells, indicating a potential role for STEF-mediated Rac1 activity in the regulation of NMMIIB.

To analyse whether a reduction in nuclear stiffness and myosin-generated tension at the nuclear envelope might impact on mechanotransduction pathways, we investigated the effect of STEF depletion on the activity of the Transcriptional Coactivator with PDZ-binding Motif (TAZ), a key transcriptional mediator of mechanical force[47]. YAP/TAZ mediated regulation of gene expression can be force-regulated, and has been demonstrated to be dependent on actin stress fibre formation and actomyosin contractility[47–49]. To investigate whether the impairment of the contractile perinuclear actin cap in STEF-depleted cells influences the transcriptional activity of TAZ, we measured the expression levels of two well-characterised TAZ-regulated genes, Ctgf and Cyr61, by qRT-PCR in control and STEF-depleted MEFs. We observed a substantial down-regulation of both Ctgf and Cyr61 in STEF KO MEFs (STEF KO DMSO) in comparison to control MEFs (Control DMSO) (Fig. 6g and Supplementary Fig. 6d). Furthermore, this observed down-regulation of TAZ target genes in STEF KO MEFs was equivalent to or greater than the effect of treating control MEFs with ML7 as a positive control (Fig. 6g and Supplementary Fig. 6d). We subsequently observed similar changes in vivo, as Ctgf expression levels were substantially lower in liver tissue isolated from STEF KO mice compared with WT mice (Fig. 6h). Taken together, these results indicate that the loss of STEF and consequent disruption of the actin cap may impact on mechanosensing signalling pathways and alter gene expression.

## Discussion

Our study provides important new insights linking the spatial regulation of Rac1 activation and signalling with the functioning of the perinuclear actin cap, a cytoskeletal organelle important for regulating nuclear shape and positioning, and cellular processes including directed migration and mechanotransduction (see model in Fig. 7).

Localised small GTPase signalling is important in cell migration. TIAM1-driven Rac1 activation at the leading edge underpins lamellipodial protrusion[32,50]. However, Rac1 is not restricted to the leading edge; it is known to also localise to the nuclear envelope, an association regulated by palmitoylation[20], with a prominent activation of juxtanuclear Rac1 seen in migrating fibroblasts[23]. Our work is the first to our knowledge to uncover a nuclear-envelope localised GEF for Rac1. STEF/TIAM2 is comparatively understudied among Rho GEFs. Previous studies have primarily focussed on the role of STEF-mediated Rac1 activation in neuronal development and physiology, while our previous work showed that STEF regulates focal adhesion disassembly[38,51]. In this study we found a prominent perinuclear localisation of STEF in a number of different cell types, where it co-localises with two key perinuclear proteins linked to the regulation of the actin cap: Nesprin-2G and NMMIIB. Interestingly, this localisation of STEF to the nuclear envelope is not shared by its homologue TIAM1, despite sharing a similar overall domain structure and significant sequence homology in the catalytic domain[49]. However, the C-terminal PH domain required for perinuclear STEF localisation shares only 54% sequence identity with the homologous domain of TIAM1, which may explain their differential localisation.

Regulation of the nucleus is emerging as an increasingly important aspect of cell migration; there is mounting evidence that control of the orientation, positioning, morphology and mechanical properties of the nucleus is critical for efficient cell migration, particularly in a 3D context where the cell has to navigate the confines of the extracellular matrix[1,5,10,11,27]. Forces acting on the nucleus via the actin cap are thought to control these processes. In both MEFs and polarised, migrating U2OS cells, we observed that loss of STEF-regulated Rac1 activity resulted in a profound reduction of actin cap cables, while basal actin structures were unaffected, suggesting that the localised pool of Rac1 generated at the nuclear envelope specifically regulates the actin cap. Proposed roles of the actin cap include limiting rotation of the interphase nucleus[1,7–10], constraining nuclear shape[2] and mechanotransduction[5,6,12]. In keeping, STEF-depleted cells failed to re-orientate their nuclei with the cellular axis of migration, displayed increased nuclear height and had softer nuclei (which was not due to a perturbation in lamin expression). In addition, STEF-depleted cells displayed a reduction in TAZ-regulated gene expression in vitro and in vivo. Although we observed an alteration of TAZ activity in STEF-depleted cells, we did not observe an effect on the nucleo-cytoplasmic shuttling of TAZ, in contrast to our previous studies on TIAM1[52], suggesting a different mechanism of action of TAZ regulation.

Specifically restoring Rac1 activity at the nuclear envelope in the absence of STEF was sufficient to rescue the effects of STEF-depletion on perinuclear actin cables, confirming that disruption of the actin cap upon depletion of STEF is indeed a Rac1 dependent phenotype. Previous studies investigating the formation of the actin cap indicated a crucial role for RhoA activity in the generation of perinuclear actin stress fibres and ROCK-mediated regulation of actomyosin contractility; our study does

**Fig. 4** Rac1 activity at the nuclear envelope regulates the actin cap. **a** Representative spinning disc confocal images of live parental U2OS and STEF-depleted CRISPR clones (CRISPR#1 and CRISPR #2) expressing the Raichu-Rac1 FRET probe, showing YFP signal intensity, FRET intensity as calculated from CFP/YFP and the FRET intensity in an isolated region of interest (ROI) around the nuclear envelope. Scale bar = 10 μm. **b** Quantification of the FRET ratio in the perinuclear region of parental U2OS and STEF-depleted CRISPR clones (CRISPR #1 and CRISPR #2). Data shown are pooled from three independent replicates. Statistical significance was verified using a one-way ANOVA, using Dunnett's multiple comparison test to compare the means of the CRISPR #1 and CRISPR #2 to the control. **** $p < 0.0001$. **c** Schematic representation of the structure of the eGFP-Rac1-KASH-Ext construct and its localisation at the nuclear envelope. **d** Representative confocal images of control MEFs expressing a GFP-only vector (top row) and STEF KO MEFs expressing either a GFP-only vector (second row), nuclear envelope localised constitutively active Rac1, eGFP-V12-Rac1-KASHext (third row), or nuclear envelope localised WT Rac1, eGFP-WT-Rac1-KASHext (fourth row). All MEFs are stained for DNA (Hoechst), GFP and F-actin (Phalloidin). Scale bar = 10 μm. **e** Quantification of apical actin cable number over nuclear area from cells as in **d**. Values represent the mean of three independent experiments ( >10 cells per condition, per replicate). Statistical significance was verified using a one-way ANOVA, using an uncorrected Fisher's LSD multiple comparison test to compare the means of each sample. * $p < 0.05$, n.s. = not significant. **f** Representative widefield immunofluorescence images of control MEFs either untransfected (top row) or expressing nuclear envelope localised dominant negative Rac1, eGFP-N17-Rac1-KASHext (bottom row), stained for DNA (Hoechst), GFP and F-actin (Phalloidin). Scale bar = 5 μm. **g** Quantification of apical actin cable number over nuclear area from cells as in **f**. Values represent the mean of three independent experiments ( > 15 cells per condition, per replicate). Statistical significance was verified using a paired t-test. * $p < 0.05$. Error bars represent S.E.M. throughout

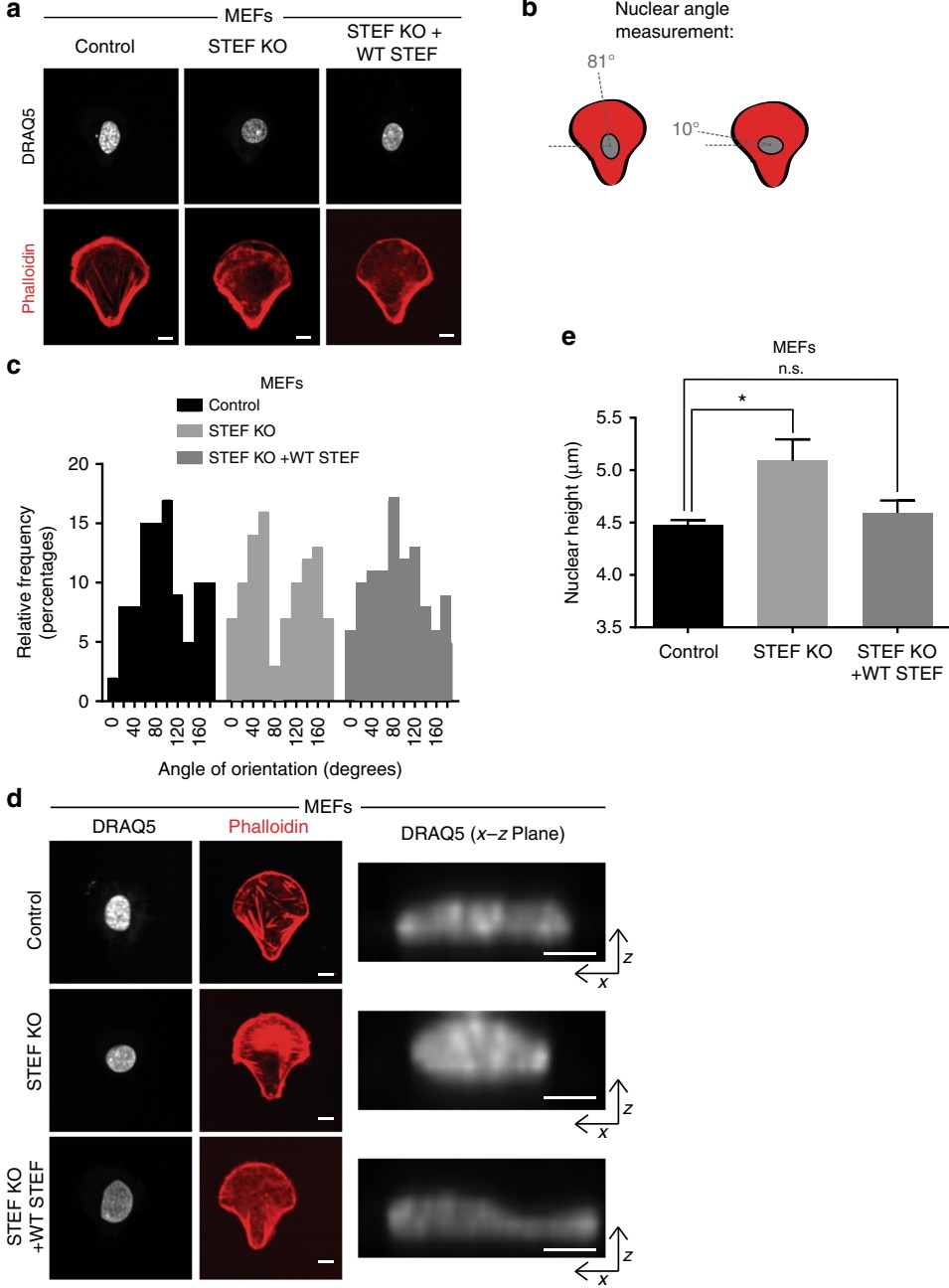

**Fig. 5** STEF regulates nuclear re-orientation and height. **a** Spinning disc confocal images of either control MEFs (Control), or MEFs depleted for endogenous STEF (STEF KO) or depleted for endogenous STEF, but re-expressing exogenous wild-type STEF (STEF KO + WT STEF) plated on collagen-coated cross-bow micropatterns, fixed after 7 h and stained for nuclei (DRAQ5) and actin (Phalloidin). **b** Schematic of the front-rear polarised morphology and nuclear position on micropatterns also showing how nuclear angles were measured. **c** Histogram showing quantification of nuclear angles relative to x axis in MEFs of **a**. Representative replicate from two independent experiments. Analysis conducted using the 'fit ellipse' tool on ImageJ. **d** Representative immunofluorescence images of MEFs plated as in **a**. Right panels, maximal projection images of the x–z plane of the nucleus. **e** Quantification of nuclear height of MEFs of **d**. Image z-stacks for DRAQ5 stain were imported into the Imaris software and the shortest principal axis of the nucleus (height) was measured. Values represent the mean of three independent experiments, >100 cells per condition, per replicate. Statistical significance was verified using a one-way ANOVA, using Dunnett's multiple comparison test to compare the means of each sample to the control. * $p < 0.05$, n.s. = not significant. Error bars represent S.E.M. Scale bars = 10 μm throughout

not contradict these data. While we do not see global changes in RhoA activation in STEF-depleted cells, we hypothesise that integrated signalling of Rac1 and RhoA GTPases may be required for orchestrating the formation and maintenance of the actin cap. Our previous work has shown a role for GEFs in directing the specificity of downstream Rac1 activity through scaffolding of protein complexes that interact with Rac1[53]. Given the ability of

perinuclear targeted V12Rac1 to compensate for STEF depletion, potential scaffolding roles of STEF seem less relevant to regulating nuclear shape and position than STEF's ability to produce a pool of active Rac1 in the proximity of perinuclear effectors that regulate the actin cytoskeleton and contractility. Interestingly, targeting of WT Rac1 to the nuclear envelope in the absence of STEF did not rescue the actin cap phenotype. This result

highlights the absence of other Rac1 GEFs at the nuclear envelope that can functionally compensate for STEF.

The exact role of STEF and Rac1 in actin cap regulation remains to be uncovered; we cannot as yet distinguish between a role for Rac1 in the formation of the actin cap vs. stabilisation of the actin cap vs. attachment of preformed actin cables to the nuclear envelope. Nonetheless, a mechanism involving the regulation of perinuclear myosin activity seems probable. There is growing evidence to support a role for Rac1 in the regulation of actomyosin-driven contractility[35–37,53]. We show a substantial

loss of myosin activity upon STEF-depletion, as evidenced by a reduction in both perinuclear pMLC staining as well as a reduction in tensile forces at the nuclear envelope. Many of the downstream phenotypes we observe are strikingly similar to those observed following down-regulation of NMMIIB, again implying that STEF-regulated Rac1 activity likely impacts on NMMIIB activation. Rac1 could potentially regulate NMMIIB activity via PAK1-mediated phosphorylation[36] or through inhibition of myosin light- chain phosphatase[37]. Further experiments will determine whether Rac1 directly regulates NMMIIB activity,

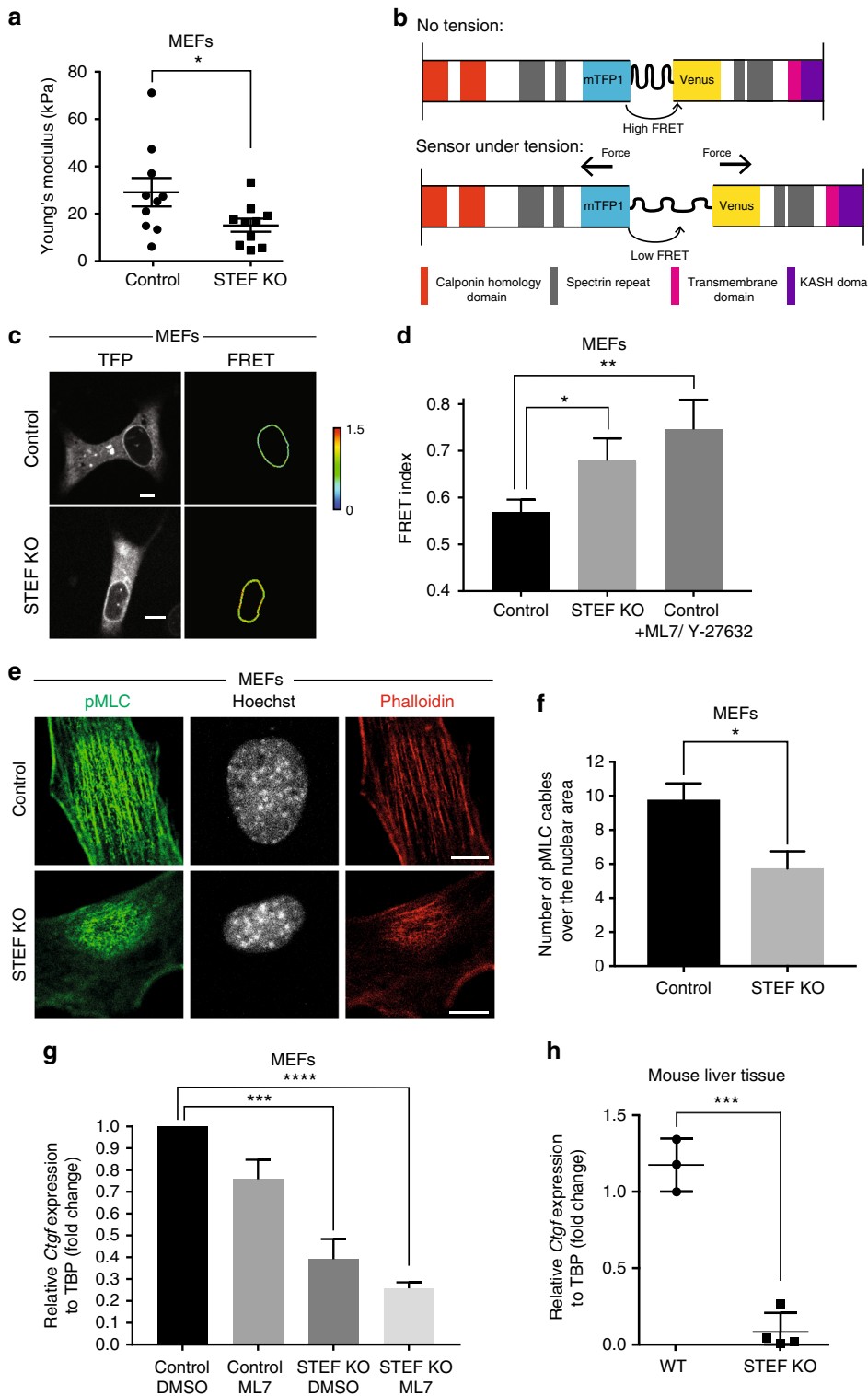

which in turn affects the actin cap, or whether a separate STEF/Rac1 pathway affects NMMIIB indirectly through regulating actin polymerisation and bundling.

Our work highlights once more how spatio-temporal regulation of the Rho GTPase family of proteins is vital for a multitude of cellular processes. Our findings add to the growing literature on the actin cap by adding an additional layer of regulation by which the cell can modulate perinuclear actin dynamics. We hypothesise that STEF regulation of Rac1 activity is a mechanism by which cells can dynamically alter their nuclear properties, such as during 3D migration where cells need to respond to a complex spatial environment, and will therefore likely be important in normal and malignant cell migration.

## Methods

**Plasmids and cloning**. Details of plasmids used in this study are summarised in Supplementary Table 1.

**Antibodies**. Details of primary antibodies, secondary antibodies and fluorescent stains used in this publication are to be found in Supplementary Table 2.

**Primers**. Sequences of all primers used in this project are to be found in Supplementary Table 3.

**Synthesis of eGFP-Rac1–KASHext constructs**. We generated DNA sequence which included: (1) the C-terminus of Rac1 (the entire 3′ region from an internal PvuII restriction site to the penultimate codon), (2) codons for the last 65 amino acids of WT human Nesprin-2a and (3) codons derived from the reverse transcription of the 'ext' amino acids—VDGTAGPGSTGSR, with addition of a STOP codon and an EcoRI site for cloning into our eGFP-WT-Rac1 vector in pcDNA3.1.

This construct was synthesised by Genewiz and inserted into pcDNA3.1-eGFP-WT-Rac1. The complete synthesised sequence was:

ACGATCGAGAAACTGAAGGAGAAGAAGCTGACTCCCATCACCTATC
CGCCAGGGTCTAGCCATGGCTAAGGAGATTGGTGCCGTAAAATACCTGGA
GTGCTCGGCGCTCACACAGCGAGGCCTCAAGACAGTGTTTGACGAAGCG
ATCCGAGCAGTCCTCTGCCCGCCTCCCGTGAAGAAGAGGAAGAGAAA
ATGCCTGCTGTTGAGCACCCGCCCGCAGCGCAGCTTTCTGAGCCGCGT
GGTGCGCGCGGCGCTGCCGCTGCAGCTGCTGCTGCTGCTGCTGCTG
CTGGCGTGCCTGCTGCCGAGCAGCGAAGAAGATTATAGCTGCACCCAG
GCGAACAACTTTGCGCGCAGCTTTTATCCGATGCTGCGCTATACCAAC
GGCCCGCCGCCGACCGTGGATGGCACCGCGGGCCCGGGCAGCACCGGC
AGCCGCTAGGAATTC

The entire sequence of the tagged Rac1 construct was confirmed by sequencing. The WT-Rac1 was then mutated to the V12 and N17 variants by QuickChange site-directed mutagenesis, and confirmed by sequencing.

**Cell culture**. All cell lines were cultured in a humidified incubator (5% $CO_2$; 37 °C). MEFs (generated in-house), U2OS (from ECACC), A549, COS-7, and virus-producing Phoenix GP cell lines (from CRUK MI sources) were maintained in Dulbecco's Modified Eagle's Medium (DMEM AQ Medium) (Gibco), supplemented with 10% Tetracycline-free Fetal Bovine Serum (FBS) (Gibco). For antibiotic selection, cell lines harbouring the pRETROX-Tet-On were grown in media supplemented with 750 μg/mL G418 (Sigma-Aldrich). Cell lines harbouring both the pRETROX-Tet-on and pRETROX-Tight-Pur were grown in media

supplemented with 750 μg/mL G418 (Sigma-Aldrich) and 2 μg/mL Puromycin (Sigma-Aldrich). Cells were regularly checked for mycoplasma infection through in-house facilities.

**Generation of cell lines**. Reverse transfections were performed for all cell lines. Transfection of plasmid DNA was performed using either FugeneHD® transfection reagent (Roche) or Lipofectamine® LTX with Plus™ transfection reagent (Thermo Fisher Scientific), according to manufacturer's protocol. For inducible over-expression, MEFs were retrovirally transduced (as described in[54]) with pRETRO-Tet-On (Clontech) followed by selection with 750 μg/mL G418 (Sigma-Aldrich). Cells were then further retrovirally transduced with a STEF expression construct (pRETROX-Tight-Pur-STEF-HALO/ pRETROX-Tight-Pur-STEF-DH*-HALO or other mutant constructs) and selected using 2 μg/mL Puromycin (Sigma-Aldrich).

To generate STEF-depleted CRISPR clones, the CRISPR design tool (http://tools.genome-engineering.org) was used to design appropriate sgRNA oligonucleotides targeting exon 1 of the human STEF sequence. Sequences were selected with low predicted off-target effects and the forward and reverse oligonucleotides were generated by MWG operon.

The sequence of the sgRNA used for resultant CRISPR clones in U2OS is outlined below:

CRISPR sgRNA FOR: 5′-CACCGCCACCGAGTCTCGATGCGTA-3′
CRISPR sgRNA REV: 5′-AAACTACGCATCGAGACTCGGTGGC-3′

Manufactured sgRNA oligonucleotides were resuspended in nuclease-free water to a final concentration of 100 μM. A mixture was prepared containing 1 μL of each sgRNA oligonucleotide at 100 μM, 1 μL of 10 × T4 ligation buffer, 1 μL of T4 PNK (Polynucleotide Kinase), made up in nuclease-free water to a total volume of 10 μL. Phosphorylation and annealing of oligonucleotides was conducted in a thermocycler using the following protocol; 37 °C for 30 min, 95 °C for 5 min and a ramp down to 25 °C at 5 °C per min intervals. The annealed and phosphorylated oligonucleotides were diluted at 1:200 with nuclease-free water and ligated into the pSpCas9(BB)-2A-GFP vector. The resultant plasmid was verified by sequencing from the U6 promoter using the U6-Fwd sequencing primer. CRISPR single cell clones were generated in the human osteosarcoma cell line, U2OS. U2OS cells were reverse transfected with the pSpCas9(BB)-2A-GFP vector containing sgRNA inserts using Fugene6 transfection reagent, according to the manufacturer's protocol. Fluorescence activated cell sorting (FACS) was used to select the GFP-positive cell population and transfer single cells to individual wells of a 96-well plate. Single-cell clones were grown up for subsequent screening.

Further validation of selected CRISPR clones was conducted to detect the presence of indels using the SURVEYOR nuclease assay. Genomic DNA was extracted from parental U2OS cells and selected CRISPR clones, and a 1 kb region around the CRISPR target was amplified by PCR. The PCR product was purified using the QIAQuick PCR purification kit (Qiagen), according to the manufacturer's protocol, and a mixture of the diluted elution product and 10 × Taq polymerase buffer was prepared. The mixture was annealed in a thermocycler using the protocol outlined in[55], to generate DNA heteroduplexes. The annealed heteroduplex was then prepared in a mixture with 10 × NEB buffer #2, T7 endonuclease and nuclease-free water and incubated at 37 °C for 20 min. Samples were prepared and run on a 2%, 1 × agarose/TAE gel.

**Generation of MEFs**. The conditional STEF knockout mouse was designed and engineered by the 'Transgenic Technology' laboratory at the CRUK Beatson Institute, Glasgow under National Home Office guidelines and experiments were approved by the ethical review body of Glasgow University. MEFs were extracted from embryos of embryonic day 10 (the sex of the embryos was not determined). Genomic PCR was performed from these embryos to determine their genotype. Early passage, primary MEFs were immortalised using an SV40 large t-antigen expression vector, kindly provided by the Cell Regulation laboratory at the CRUK MI. MEFs were plated at a density of $1 \times 10^5$ cells per well in a 6-well plate and

**Fig. 6** STEF regulates myosin-generated tension on the nuclear envelope. **a** Atomic force microscopy measurement of nuclear stiffness (calculated as mean Young's modulus) in Control and STEF KO MEFs (10 cells per condition). Representative replicate from two independent experiments. Statistical significance was verified using an unpaired t-test, * $p < 0.05$. **b** Schematic representation of the mini-Nesprin-2G tension sensor construct (mN2G-TS). **c** Representative spinning disc confocal images of control and STEF KO MEFs expressing the mN2G-TS construct, imaged using the TFP filter to determine construct localisation, and the FRET filter to calculate the FRET Index at the nuclear membrane. **d** Quantification of mN2G-TS FRET Index at the nuclear envelope in control MEFs, STEF KO MEFs, and control MEFs after treatment with a combination of 10 μM myosin light-chain kinase inhibitor (ML7) and 10 μM ROCK inhibitor (Y-27632). Values are presented as the mean of three independent experiments. Statistical significance was verified using an unpaired t-test, * $p < 0.05$, ** $p < 0.01$. **e** Representative confocal immunofluorescence images of control and STEF KO MEFs stained for DNA (Hoechst), pMLC and F-actin (Phalloidin). **f** Quantification of apical phospho-MLC positive cables over nuclear area from control and STEF KO MEFs (~20 cells per condition, per replicate). Values represent the mean of three independent experiments. Statistical significance was verified using a paired t-test. * $p < 0.05$. **g** qPCR for the TAZ target gene Ctgf, normalised to Tbp expression in control and STEF KO MEFs pre-treated with either DMSO or 10 μM ML7 for 2 h. Data are relative to the negative control (control MEFs + DMSO) and are presented as the mean from four independent replicates. Statistical significance was verified using a one-way ANOVA, using Tukey's multiple comparison test to compare the means of each sample. *** $p < 0.001$, **** $p < 0.0001$. **h** qPCR for the TAZ target gene Ctgf, normalised to Tbp expression in liver tissue isolated from WT ($n = 3$) and STEF KO ($n = 4$) mice. Statistical significance was verified using an unpaired t-test, *** $p < 0.001$. All error bars represent S.E.M. Scale bars = 10 μm throughout

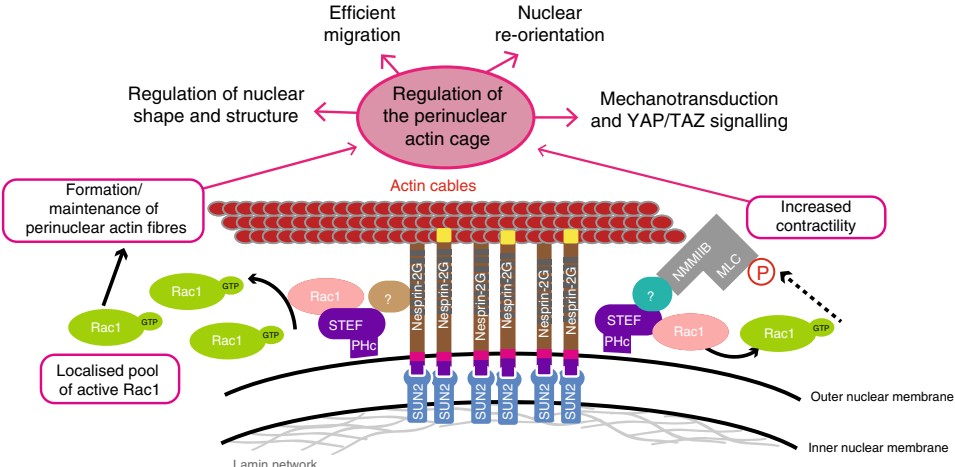

**Fig. 7** Model describing the role of perinuclear STEF. STEF at the nuclear envelope generates a localised pool of active Rac1 which is required for the formation and/or stabilisation of perinuclear actin cables and for perinuclear actomyosin contractility. Disruption of the perinuclear actin cap in STEF-depleted cells alters the structural and morphological properties of the nucleus, with consequences for directed cell migration and mechanotransduction

incubated for 24 hours before transfection with 2 μg of the SV40 expression vector using the Lipofectamine® LTX with Plus™ transfection reagent (Thermo Fisher Scientific). Culture medium was changed the following day, and when cells were 80% confluent, they were split into a 10 cm dish. Immortalised MEFs were split at 1:10 at least five times before use for functional studies as a negative selection against untransformed cells.

**Adenoviral transduction.** SV40-immortalised mouse embryonic fibroblasts (MEFs) were plated at a density of $2 \times 10^5$ cells per 10 cm dish. The following afternoon, cells were washed once with $PBS^{-/-}$ and 5 mL of 0.2% serum culture medium was added. 5 μL of adenoviral-Cre-GFP (University of Iowa) or an empty adenoviral-GFP control (University of Iowa) was added to the medium and cells were incubated overnight. The following morning, medium was replaced with fresh, 10% serum culture medium and cells were allowed to recover. MEFs were cultured for a further 72 hours post-infection before use in functional studies, to allow time for recombination and protein degradation.

**Protein analysis.** Cells were lysed in an appropriate volume of IP lysis buffer [50 mM Tris-HCl pH 7.5, 150 mM NaCl, 1% Triton-x-100 (v/v), 10% glycerol (v/v), 2 mM EDTA, 25 mM NaF and 2 mM $NaH_2PO_4$ containing 1% protease inhibitor cocktail (Sigma-Aldrich) and 1% phosphatase inhibitor cocktails 1 and 2 (Sigma-Aldrich) added fresh] or RIPA buffer [25 mM Tris pH 7.5, 150 mM NaCl, 0.1% SDS (v/v), 0.5% sodium deoxycholate (v/v), 1% Triton-x-100 containing 1 EDTA-free protease inhibitor tablet (Roche) and 1% phosphatase inhibitor cocktails 1 and 2 (Sigma-Aldrich) added fresh] for 10 min on ice and proteins were resolved by SDS-PAGE for western blotting. For immunoprecipitation, lysates were pre-cleared overnight at 4 °C with an appropriate IgG pre-bound to 50 μL of GammaBind G Sepharose beads (Amersham). After pre-clearing, lysates were incubated with the appropriate antibody/IgG control pre-bound to 25 μL of GammaBind G Sepharose beads (Amersham, blocked with 5% BSA overnight at 4 °C), for 4 hours at 4 °C. Beads were subsequently washed with RIPA buffer, and eluted with 1 × SDS-PAGE sample buffer (Nupage, Invitrogen). For in vitro pull-down assays, Nesprin-2G, NMMIIB or PAR3 protein was immunoprecipitated from U2OS cell lysates as described above. Beads were washed with a high salt solution to remove any intermediary proteins and were incubated with purified, recombinant STEF protein for 1 hour at 4 °C. Beads were subsequently washed with RIPA buffer, and eluted with 1 × SDS-PAGE sample buffer (Nupage, Invitrogen). Full scans of all blots can be found in Supplementary Fig. 7.

**GTPase activity assay.** Small GTPase protein activity assays were conducted with control and STEF KO MEFs using the G-LISA® kit (Cytoskeleton) following the manufacturer's instructions.

**Biochemical fractionation.** Fractionation to enrich for perinuclear proteins was performed as in ref [24]. U2OS or MEF cell pellets were lysed in an appropriate volume of perinuclear-enriched extraction buffer A (40 mM HEPES pH 7.4, 120 mM KCl, 2 mM EGTA, 0.5% Glycerol, 10 mM beta-glycerophosphate, and 0.5% NP-40 in ddH$_2$O) for 30 min on ice. Lysates were centrifuged at $500 \times g$ for 5 min and the supernatant was removed as the cytoplasmic fraction. The pellet was washed once in perinuclear-enriched extraction buffer A (minus NP40) and then lysed in an appropriate volume of perinuclear-enriched extraction buffer B (10 mM Tris-HCl pH 7.4, 1.5 mM KCl, 0.5% Triton X-100, 0.5% deoxycholate in ddH$_2$O

containing 2.5 mM MgCl$_2$ and 0.2 M LiCl added fresh) for 45 min at 4 °C with rotation. Lysates were centrifuged at $2000 \times g$ for 5 min and the supernatant was removed as the perinuclear-enriched fraction. The pellet was washed once in perinuclear-enriched extraction buffer B and the remaining pellet was dissolved in an appropriate volume of perinuclear-enriched extraction buffer C (8 M Urea in ddH$_2$O) and centrifuged at $10 \times g$ for 10 min, to yield the core-nuclear fraction. Finally, all resultant lysate fractions were centrifuged at $15871 \times g$ for 5 min before preparation of lysates in 1 × SDS-PAGE sample buffer (Nupage, Invitrogen) to resolve proteins by SDS-PAGE and western blotting.

**Immunofluorescence.** For immunofluorescence, cells were grown on coverslips and fixed with either 100% ice-cold methanol for 5 min at −20 °C or 3.7% formaldehyde for 15 min at room temperature. Cells were permeabilised for 3 min in 0.2% Triton in PBS$^{-/-}$ (v/v), washed and then blocked in 1% BSA in PBS (v/v) for 1 hour, before successive incubation with primary and then secondary antibodies. Coverslips were mounted onto glass slides using a droplet of ProLong® Gold anti-fade reagent containing the DNA stain DAPI. For digitonin permeabilisation, formaldehyde fixed cells were permeabilised for 3 min in 40 μg/mL digitonin in PBS (v/v), and stained as outlined above.

**Microscopy.** Images were obtained using a variety of microscopes. Immuno-fluorescence imaging of STEF localisation were captured on the Leica GatedSTED SP8 (gSTED) microscope system. This system utilises LAS AF Lite software (Leica) to capture and process images. Images were taken using the 100 × or 60 × oil lens. Immunofluorescence images of actin cables and pMLC were captured using the Deltavision Core system(based on an Olympus IX71 microscope; fluorescence is achieved using a 300 W Xenon light source with a variety of Sedat filter sets (406, 488, 568, 647 nm) and the attached Roper Cascade 512B camera; images were taken using 100 × / 60 × oil lens.), or the Zeiss LSM800 Airyscan, with a Zeiss Observer. Z1 body and Zeiss 63 × / 1.4 NA oil Plan-Apochromat lens. The Deltavision core system utilises softWorx to capture and process images. Images of micropatterned cells were obtained using the Opera Phenix™ High Content Screening System, which is a microlens enhanced Nipkow spinning-disk confocal microscope with four sCMOS cameras for simultaneous four-channel image capture, with temperature and CO$_2$ control to support live cell imaging. The system utilises the Harmony software for image capture and processing, with further capacity for image analysis in the Columbus™ Image Data Storage and Analysis System. FRET experiments with the mN2G-TS construct were conducted using the 3i Marianas microscope system, with attached Evolve EMCCD camera. All other images were captured using a Zeiss Axiovert 200 M microscope (Solent Scientific). The system uses an Andor iXon 888 camera and a 300 W Xenon light source is used for fluorescence illumination with a variety of ET-Sedat filters (406, 488, 568, 647 nm). The system utilises the Metamorph software to capture and process images. Images were taken using the 100 × oil lens.

**Duolink PLA.** Duolink PLA was performed using the Duolink II Red Starter Kits (Sigma) following the manufacturer's instructions. A video summarising the steps of this technique can be found online (www.olink.com/products-services/duolink/howuseduolink). In summary, cells were seeded at a density of $2 \times 10^5$ cells per well of a 6- well plate, and fixed in 3.7% formaldehyde after 48 hours. Cells were permeabilised, blocked and stained with appropriate primary antibody solutions, as described above. Following incubation with primary antibodies, coverslips were

washed and incubated with a mixture of the PLUS/MINUS Duolink® PLA probes (OlinkBioscience) for 1 hour at 37 °C. Coverslips were then washed and detection of signal was conducted using the Duolink® detection reagent kit (red) (Olink-Bioscience). Coverslips were first incubated with a ligation-ligase mixture for 30 min at 37 °C. Coverslips were subsequently washed and further incubated with an amplification-polymerase mixture for 100 min at 37 °C. Final washing steps were conducted; coverslips were first washed twice in Duolink Buffer B for 10 min, followed by a 1 min wash in a 0.01% diluted Duolink Buffer B (v/v dilution in ddH$_2$O). Coverslips were left to air-dry, protected from light, before mounting onto glass slides with ProLong® Gold anti-fade reagent containing the DNA stain DAPI. Coverslips were imaged using the Zeiss microscope system; at least 10 images were taken per condition. Duolink® signal in immunofluorescence images was quantified using the Cell Profiler Software. A pipeline was designed to count fluorescent speckle number within the nuclear area, delineated by the DNA stain DAPI.

**Micropattern experiments**. CYTOOplates™ 96 RW-CW-M-A (CYTOO) were utilised for all experiments; the 96-well plate format had glass-bottomed wells with medium sized (1100 μm$^2$), crossbow-shaped, activated micropatterns. Prior to the assay, micropattern plates were coated with 20 μg/mL rat tail collagen type I (Corning), diluted in sterile PBS. Micropattern plates were pre-warmed at 37 °C during the preparation of cells for plating. MEFs had been pre-cultured in the presence or absence of doxycycline for 24 hours before infection with adenovirus as outlined previously. 3200 cells were plated per well of the 96-well plate, with gentle agitation to prevent cell clumping and cells were incubated at 37 °C for 7 hours before fixation in 3.7% formaldehyde and antibody staining as described previously. DRAQ5 and Phalloidin dyes were used to stain the nucleus and actin cytoskeleton respectively. Z-stacks were captured at 1 μm steps, across a range of 10 μm. Micropatterns occupied by a single cell, with a single nucleus, with the front-rear polarised morphology were analysed. To measure nuclear re-orientation, the middle plane of the stacks was used and images were exported to ImageJ software. The 'fit ellipse' tool was implemented on the DRAQ5 channel image to calculate the angle of the nucleus, relative to the horizontal axis. For nuclear height analysis, stacks were exported to the Imaris (Bitplane) software using the 'cells' image processing pipeline on the DRAQ5 channel to measure the 3D height of the nucleus. This was defined by the software as 'BoundingBoxOO Length A—the shortest principal axis of the nucleus'.

Cells were manually selected for analysis according to the following criteria; cells must be occupying the set micropattern position, cells must have only a single nucleus (with no secondary punctate staining of DRAQ5 in the cell), cells must have fully adopted the polarised morphology (Phalloidin staining used to verify this), the nucleus must be completely contained within the boundary of the Phalloidin staining. These criteria were applied to cells of all treatment conditions prior to analysis.

**FRET imaging of tension sensor**. The nuclear force tension sensor (mN2G-TS) was synthesised as a fusion construct of the tension sensor module (https://www.addgene.org/26021/)[46] and human mini Nesprin-2 construct (Nesprin delta460–6643)[56]. The tension sensor module was inserted after aa460 of the Nesprin-2 construct. During imaging, fluorescence bleed-through and lifetimes were controlled using an equivalent construct (mN2G-TFP), lacking only the venus sequence. Images were taken on a 3i Marianas spinning disk confocal system using 445 nm, 515 nm lasers and 482/35, 542/12 single band-pass emission filters for TFP and Venus detection, respectively. FRET values were calculated using two channel sensitised emission FRET in slidebook (3i) software as described by the manufacturer. To assess nuclear envelope FRET values only, a single pixel-wide mask was drawn on the nuclear envelope.

**FRET imaging of Rac1 biosensor**. Sparsely-seeded U2OS cells expressing the Raichu-Rac1-KRas-CFP-YFP construct were imaged using a 3i spinning disk inverted confocal microscope with a Zeiss ObserverZ microscope frame and using a Plan-Neofluar 100 × /1.3 objective. Samples were excited using 440 and 488 nm diode lasers via Zeiss CFP, YFP and CFP/YFP filter cubes respectively. Images were collected using a Photometrics Evolve EMCCD camera. Single confocal planes were then analysed manually using ImageJ. After application of a Gaussian blur of 0.5 pixels to aid pixel registration between channels, the CFP image was divided by the YFP image using a 32-bit floating point calculation to generate the FRET ratio image. A region of interest 2 pixels wide was drawn around the nucleus (as discerned from the YFP image), and this region was then used to measure the average FRET intensity from the FRET ratio image.

**Atomic force microscopy**. AFM experiments were conducted using an AFM microscope system at the Bioimaging facility, The University of Manchester. MEFs were plated onto glass bottomed 35 mm dishes and mounted onto the inverted microscope in media. A silicon nitride, quadratic pyramid cantilever probe was used at angle 0°. The cantilever was positioned above the nucleus and was descended towards the nucleus at a speed of 2 μm/s. At least 15 force–distance curves were collected per cell, and at least 10 cells were measured per condition. The force–distance curves were analysed using the JPK Imaging Processing software to obtain Young's modulus values.

**qRT-PCR**. RNA was extracted from cell pellets or tissue using the RNeasy kit (Qiagen) with additional digestion of genomic DNA with RNase-free DNase kit (Qiagen), according to the manufacturer's instructions. 1 μg of RNA was reverse transcribed to cDNA using the Omniscript RT Kit (Qiagen) according to the manufacturer's instructions. qPCR was performed in quintuplicates in a 10 μL reaction mixture containing 5 μL of 2 × TaqMan® master mix, 0.5 μM of each of the primers, 0.1 μL of the appropriate Universal probe and 10 ng of cDNA. *Tbp* expression levels were used to normalise for differences in RNA input. Primers for *Ctgf*, *Cyr61* and *Tbp* were designed online using the Universal ProbeLibrary Assay Design Center (Roche). For analysis from mouse livers, livers were obtained from 18.5 day embryos from STEF total knockout mice in C57/BL6 background (the sex of the embryos was not determined).

**Statistical analysis**. Appropriate statistical tests were chosen to minimise type I error associated with significance values. Statistical differences between data were analysed in Prism (GraphPad Software) with either an unpaired, two-tailed Student's t-test or an unpaired, one-way ANOVA, with appropriate post hoc multiple comparisons test. Tests are specified in figure legends along with *P* value significance.

**Data availability**. The authors declare that all data supporting the findings of this study are available within the article and its supplementary information files or from the corresponding author upon reasonable request.

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

## Acknowledgements

The authors thank members of the Cell Signalling lab for their critical reading of the manuscript and Lynsey Hurley for her helpful guidance. We thank the Molecular Biology Core Facility and Advanced Imaging Facility of Cancer Research UK Manchester Institute (CRUK MI) for their assistance with sequencing, qRT-PCR and microscopy and also the BioImaging Facility at The University of Manchester for the use of their microscope suite and assistance with the AFM experiments, in particular Steven Marsden. This work was supported by Cancer Research UK (grant number C5759/A12328), MRC (grant number MR/L007495/1) and Worldwide Cancer Research (WCR 16–0379). Work in the laboratory of K.M.H. is supported by the NIH, grant number: GM-R35GM122596. We also thank The University of Manchester who have provided us with lab space, reagents and equipment following the fire at the Paterson Building, and all those who kindly provided us with constructs, reagents and protocols for the project, particularly the Hoshino lab.

## Author contributions

A.W. performed the majority of the experiments, data analysis, and manuscript preparation. A.P. performed the Rac1 FRET, digitonin permeabilisation, STEF deletion mutant localisation and Rac1 targeting experiments in collaboration with D.T.N. and edited the manuscript. G.W. isolated MEFs from conditional STEF knockout mice and genotyped them, performed most of the cloning for the project and prepared cell cultures for several experiments for immunofluorescence. D.T.N. performed the Rac1 targeting experiment and pMLC analysis. Z.D. performed the Lamin expression analysis and qPCR analysis of *Ctgf* expression in mouse tissue. T.W. contributed to data analysis. C.R. first identified the perinuclear localisation of STEF. D.S. generated the conditional STEF knockout mice in collaboration with O.J.S. D.J.M. performed pilot FRET experiments and analysis under the supervision of K.M.H. T.Z performed the FRET tension sensor experiments and also supervised D.T.N. and T.W. A.M. was the grant holder and principal investigator who supervised the study, edited the manuscript and made intellectual contributions throughout.

## Additional information

**Competing interests:** The authors declare no competing interests.

