## [Peer Review File · Nature Communications]

Reviewers' comments:

Reviewer #1 (Remarks to the Author):

T-lymphoma invasion and metastasis-inducing protein 2 (STEF/ Tiam2) is a relatively poorly characterized GEF that is highly expressed in the brain. Like Tiam 1, this protein contains an N-terminal N-myristoylation signal that allows it to associate with membranes, as well as PH domains which also serve as selective membrane anchors.

This MS shows that endogenous & C-terminal tagged STEF appears to be nuclear envelope enriched. This would not only provide a means to activate Rac1/2/3 at this site, but might suggest it could play a role in locally regulating F-actin. This in turn could allow organization or coupling between the apical actin cap and the plasma membrane & regulate nuclear morphology and re-orientation during front-rear polarisation. However Rac1 is not associated with actin stress fibre (SF) production and therefore these parts of the study need to be assessed more carefully.

The authors do provide evidence that myosin-II generated tension at the nuclear envelope is altered, using a sensor in the MEF STEF KO cells. However tension through conventional actin stress fibres is thought of as being downstream of RhoA-ROCK. Finally the connection between Cdc42 and cell polarization is clearly established (via polarization molecules and the kinase MRCK) however the link between Rac1 and polarization is much less clear.

Overall this is an interesting and important study. However the authors convince us as to how STEF is nuclear envelope localized, whether this is isoform specific (ie versus Taim1) and whether its presence at this site leads to elevated Rac1.GTP at this site (this use of FRET would be interesting). The tension sensor is less so.

In addition the authors should consider the issues as outlined below.

1. Using an anti-body and a single allele KO of STEF in U2OS cells the authors provide evidence that STEF is enriched at the nuclear envelope. It would be important for the authors to also demonstrate that Tiam1 is not nuclear envelope localized (although it is sequence related, and has the same mechanism of membrane attachment). Therefore they should carry out proper structure/ function analysis of this nuclear localization in STEF. Based on their previous work they should have access to the required tools, and they can do this analysis in STEF KO lines. It may turn out of course that Tiam1 is nuclear-enriched also and that these genes play overlapping roles.

Suppl Fig 1C . In this regard the supplementary data on the FL and deletion constructs should be moved to the main figures, with a clear schematic to show what is being expressed and its localization. Obviously removal of the N-myr signal by itself may be sufficient to prevent nuclear membrane enrichment, and therefore the N-terminal sequences may have to be left intact (ie perform C-terminal deletions).

2. The authors state in their abstract that STEF "interacts with the key perinuclear proteins Nesprin-2G and Non-muscle myosin IIB". However in the results section they make it clear that this is not a direct interaction, and at best one can conclude that these proteins co-localize. This should be changed.

3. In the results the authors use a condition KO of STEF in MEF to analyse the peri-nuclear actin cables (above the nucleus) in the region of the actin cap. Since Rac is of itself not associated with the production or stabilization of actin stress fibres (but rather RhoA/C), this part of the study is rather curious. The authors need to go back and look in more detail at what is going on. In particular the apical SF being shown need to be compared to the sub-nuclear SF bundles, which I think they suggest are unaffected. They can then present data for both pools (top and bottom confocal stacks/sections), showing that only one is affected.

4. Whether Rac1 is required for the stabilization of apical perinuclear actin cables is also controversial and should be tested in this study. It may be that Rac1 is required for coupling of Rho-type SF to the nucleus - and would be an important result.

5. In Figure 4d the authors appear to suggest that cells placed on crossbow matrices develop a strong sub-nuclear patch if STEF is missing (but actin SF are not clearly resolved). Given the intrinsic variability of the F-actin network multiple cells (say 10 each) need to be shown (even if only in the supplementary section).

6. Figure S5 The authors should show their anti-STEFA antibody results for MEF and MEF STEFA KO cells. The link of between loss of STEFA and nuclear-localized TAZ seems beyond the scope of this study, unless experiments in STEFA KO embryos clearly indicated a consistent change in TAZ signaling in tissues of interest that express STEFA.

Reviewer #2 (Remarks to the Author):

In this paper, Woroniuk and colleagues investigate the role of the Rac GEF STEF and show that STEF localizes at the nuclear periphery, where it regulates perinuclear actin structures. Interestingly, the authors show that STEF depletion impacts nuclear orientation and contractility-dependent transcription. The work from Woroniuk and colleagues shows some interesting data and presents an original model. However, some aspects of this study require some clarification and the molecular events downstream STEF that regulate the actin cap remain unclear.

The authors report that STEF displays perinuclear localization, however the immunofluorescence data do not allow concluding whether STEF is at the inner nuclear membrane or perinuclear. Regarding the fractionation, it is unclear what Woroniuk et al. mean by perinuclear fraction? The protocol described in the method section is a detergent-based fractionation which allows a 1st separation between a fraction (pellet) that contains the membranes (plasma and nuclear membranes)+insoluble cytoskeleton and a fraction (supernatant) that contains the cytoplasmic ("soluble") components. This does not allow specific perinuclear fractionation. Additionally, the authors used co-IP with nesprin 2G to confirm STEF localization (and STEF interactors), but nesprin 2G is over 700 kD while the figure 1e shows a band at about 300 kD. The signal here is probably unspecific or is due to another (small) isoform of nesprin 2 which may not localize at the nuclear periphery. (Similar comment can be made for Nesprin 1Giant (enaptin) which is over 1,000kD, while the authors show a band at about 250-300 kD in figure 1f) (and the co-IP in supplementary figure 2a). All these make the perinuclear localization of STEF quite questionable. Does STEF localize at the inner or at the outer nuclear membrane, or is it also nuclear, since we can also observe a nice nuclear localization of STEF on some images, such as supplementary Figure 5? The authors need to analyze more precisely STEF localization and its potential interaction with inner nuclear membrane proteins or with the lamina. The authors could also use mutant forms of STEF to identify the signaling domain that allows this peculiar localization (distinct from other related GEFs such as Tiam1, even if Tiam1 can also localize in the nucleus as shown in the supplementary figures).

While the characterization of perinuclear actin structures has thoroughly been done, it is unclear how STEF regulates these structures. The authors mentioned that STEF is a Rac GEF and the results obtained with the STEF mutant lacking the DH domain suggest that STEF regulates Rac activity at the nuclear periphery. This hypothesis is intriguing and could be tested using Rac FRET sensors to look at perinuclear Rac activity in cells depleted for STEF. If Rac is activated by STEF how does it regulate perinuclear actin cap? Does it promote NMMIIB activation (myosin phosphorylation) or does it stimulate local actin polymerization (via Arp2/3) or alternatively does it participate to the LINC complex assembly? Many pharmacological inhibitors (PAK inhibitors) or plasmid constructs could be used to further analyze the downstream effectors that regulate perinuclear actin cap formation. Can the authors recapitulate the effect by inactivating Rac (or Rac effector?) Or can they rescue STEF depletion by addressing another Rac GEF (or just the DH-PH domain) and restore Rac activity at the nuclear periphery?

Regarding the measurement of the Young's modulus using AFM in Fig 5g, the decreased nuclear stiffness could also be the consequence of an altered lamina. The authors show that STEF regulates gene expression (figure 5) and it is possible that some lamina components may be affected, resulting

in abnormal nuclear stiffness. Did the authors look at laminAC and lamin B expression and localization in cells depleted for STEF?

The authors analyzed TAZ nuclear expression (Sup figure 5) and conclude that STEF regulates TAZ activity, although usually one measures the nuclear/cytoplasmic ratio of TAZ to investigate its activity. Without doing any new experiments, the authors should re-analyze the images and see if the ratio is altered.

Minor comments:

-What do the authors mean by “nuclear morphology” in the title? Nuclear height is affected by STEF depletion, but morphology can relate to so many aspects and having it in the title could be misleading.

-Since a confocal microscope was used for imaging actin in figure 3 b and e, the authors should show a representative z projection view to allow better characterization of the perinuclear actin network (fig3b 3e)

-line 69: ref 23 –This reference may not be adequately used. In this paper published by Hahn and colleagues, we can observe perinuclear Rac-GFP, but very low perinuclear Rac activity (using FRET sensors) is reported in this paper.

-the co IP STEF/ NMMIIB is hard to distinguish since the two proteins have similar molecular weight, additional view of the uncropped gels as supplementary could be useful.

RESPONSE TO THE REVIEWERS' COMMENTS

We thank both reviewers for their critique of our study and their constructive comments on how to improve our manuscript. We respond to their comments below and present the changes we have made in the revised manuscript.

Reviewer #1

1) It would be important for the authors to also demonstrate that Tiam1 is not nuclear envelope localized...Therefore they should carry out proper structure/ function analysis of this nuclear localization in STEF.

We have expanded on the evidence provided in the original manuscript to confirm that Tiam1 does not localise to the nuclear envelope in the presence of STEF or following STEF depletion (Supplementary Fig. 1f,g). To address the structure/function relation of STEF, we have expressed a range of HA- tagged deletion mutants of STEF in U2OS cells, and examined their localisation by immunofluorescence. We found that deletion of the C-terminal PH domain abolished the perinuclear localisation of STEF, and that the C-terminal DH-PH domains were sufficient for perinuclear localisation, indicating the importance of the C-terminal PH domain in mediating the localisation of STEF to the nuclear envelope. These data are now included as Fig. 1d-f, and we have restructured the figure to include the expression images for exogenous full-length and Δ N STEF formerly included as supplementary, as suggested by the reviewer. The C-terminal PH domain of STEF has only 54% sequence homology with the corresponding domain of Tiam1, which may explain the differential localisation of the two proteins.

2) The authors state in their abstract that STEF "interacts with the key perinuclear proteins Nesprin-2G and Non-muscle myosin IIB". However in the results section they make it clear that this is not a direct interaction, and at best one conclude that these proteins co-localize. This should be changed.

We agree with the reviewer that our original wording could be misleading, therefore we have adapted the text of the paper appropriately. We suggest that these indirect interactions support the localisation of STEF at the outer nuclear envelope in close proximity to these key regulators.

3) Since Rac is of itself not associated with the production or stabilization of actin stress fibres (but rather RhoA/C), this part of the study is rather curious. The authors need to go back and look in more detail at what is going on.

We agree that this is curious, but nonetheless our data clearly implicate a role for Rac in the regulation of the actin cap. In our original submission, the involvement of Rac was implied by the inability of GEF-inactive STEF to rescue STEF-depletion effects on the actin cap. To now strengthen this association (between Rac and the actin cap), we firstly tested the effect of STEF depletion on the activity of RhoA as well as Rac1 and Cdc42 using an ELISA assay. Depletion of STEF resulted in a significant decrease in Rac1 activity, with no effect on the activity of RhoA and Cdc42 (Supplementary Fig. 4a, confirming previous reports that STEF is a Rac1-specific GEF (Hoshino *et al. JBC.* 1999, 274:17837-44, Rooney *et al. EMBO Reports.* 2010, 11:292-8). Moreover, FRET analysis using a Rac-activity biosensor revealed reduced Rac activity in the perinuclear region of STEF-depleted cells (Fig. 4a,b).

We further investigated the role of perinuclear Rac1 activity in the regulation of the actin cap by targeting various Rac1 constructs to the outer nuclear envelope (now included as Fig. 4c-g in

our revised manuscript). We show that the targeting of constitutively active-Rac1 to the nuclear envelope in STEF KO MEFs was sufficient to rescue the disruption of the perinuclear actin cap. In contrast, expression of WT-Rac1 to the nuclear envelope was not sufficient to rescue the phenotype of STEF KO MEFs, highlighting the requirement of STEF to activate Rac1 at the nuclear periphery and suggesting that no other Rac1 GEF can compensate for STEF depletion and that STEF scaffolding activity is likely not important. Additionally, expression of the dominant-negative form of Rac1 at the nuclear envelope caused significant disruption of the actin cap of WT MEFs.

We feel that these experiments confirm that disruption of the actin cap upon depletion of STEF is indeed a Rac1-dependent phenotype, and highlight the importance of STEF-mediated localised Rac1 activity at the nuclear envelope in actin cap regulation. Our study does not contradict previous studies implicating a crucial role for RhoA activity in the generation of perinuclear actin stress fibres and ROCK mediated regulation of actomyosin contractility. We propose that integrated signalling of Rac1 and RhoA GTPases orchestrate the formation and maintenance of the actin cap: possibly the formation of actin fibres in the actin cap is dominantly Rho-driven, while Rac-driven processes influence the assembly of actin fibres in the cap or engagement of the cap with the nuclear envelope. Significant in this regard is the increasing evidence for a role for Rac1 regulation of NMMIIB, which has been clearly implicated in actin cap regulation and the effects of whose depletion closely resemble STEF depletion (as detailed in the Discussion section of the manuscript). In keeping with the above statement, we demonstrate a reduction in myosin-generated tension at the nuclear envelope upon depletion of STEF, as well as a significant reduction of perinuclear pMLC in STEF-depleted MEFs, as a read-out of NMMIIB activity, now included in the manuscript as Fig. 6e,f. We feel these results strengthen our hypothesis that STEF-mediated Rac1 activity regulates the perinuclear actin cap through downstream regulation of NMMIIB activity.

4) The apical SF being shown need to be compared to the sub-nuclear SF bundles...they can then present data for both pools (top and bottom confocal stacks/sections), showing that only one is affected.

As prompted by the reviewer, we have now supplemented the data originally presented in our manuscript by additionally quantifying the number of basal sub-nuclear actin fibres, addressing the effect of STEF depletion on the basal actin network. Our analysis showed that depletion of STEF does not affect the number of basal actin cables. These data are now presented as Supplementary Fig. 3d,e, including example images showing both the basal and apical actin pools in control and STEF-depleted MEFs. Furthermore, we have included representative images of basal actin from our Rac1 targeting experiments, included as Supplementary Fig. 4e. These data confirm that STEF depletion does not have a global effect on actin organisation; we see a specific disruption of apical actin cables with no significant effect on basal actin structures.

5) Whether Rac1 is required for the stabilization of apical perinuclear actin cables is also controversial and should be tested in this study. It may be that Rac1 is required for coupling of Rho-type SF to the nucleus - and would be an important result.

The reviewer makes an excellent point. This has been addressed in our response to point 3 above.

6) In Figure 4d the authors appear to suggest that cells placed on crossbow matrices develop a strong sub-nuclear patch if STEF is missing (but actin SF are not clearly resolved). Given the intrinsic variability of the F-actin network multiple cells (say 10 each) need to be shown (even if only in the supplementary section).

We thank the reviewer for highlighting where we were unclear, and would like to clarify that we were not suggesting that depletion of STEF affects sub-nuclear actin. Rather, due to the limited resolution of these images, we cannot clearly resolve actin fibres, and therefore no analysis of actin

fibres in cells on micropatterns was conducted. We see intrinsic variability in the staining of the actin network of cells plated on micropatterns, and have now included ten example control MEFs in Supplementary Fig. 5a to demonstrate this, as suggested by the reviewer.

7) The link between loss of STEF and nuclear-localized TAZ seems beyond the scope of this study, unless experiments in STEF KO embryos clearly indicated a consistent change in TAZ signaling in tissues of interest that express STEF.

In response to this point, and to expand our work on TAZ signalling, we performed qPCR analysis of the expression levels of the TAZ-regulated gene *Ctgf* in liver tissue isolated from WT and STEF KO mice. We observed a significant reduction in *Ctgf* expression in the STEF KO samples (now included in the manuscript as Fig. 6h), consistent with the results of similar experiments performed in WT and KO MEFs (Fig. 6g and Supplementary Fig. 6e). Expression levels of *Cyr61* were too low to analyse in liver tissue of either genotype. We chose the liver as a tissue of interest as the expression of both STEF and TAZ have been linked with the progression of hepatocellular carcinoma (Chen *et al. Int J Cancer*. 2012, 130:1302-13; Xiao H *et al. Cancer Sci*. 2015, 106:151-159).

Reviewer #2

1) The authors report that STEF displays perinuclear localization, however immunofluorescence data do not allow concluding whether STEF is at the inner nuclear membrane or perinuclear. ...Does STEF localize at the inner or at the outer nuclear membrane, or is it also nuclear, since we can also observe a nice nuclear localization of STEF on some images?

As suggested by the reviewer, we have further investigated the localisation of STEF at the nuclear envelope using a selective permeabilisation experiment, now included in the revised manuscript as Fig. 1h. Digitonin permeabilisation (at the concentrations used in our study) is known to selectively permeabilise the cholesterol-rich plasma membrane, leaving the nuclear envelope intact. As such, antibodies for intra-nuclear proteins are unable to access their ligand and no signal is seen for the intranuclear proteins via immunofluorescence. We demonstrate this with our negative control lamin A/C, for which we only see a signal upon permeabilisation with 0.5% Triton which is able to permeabilise both the plasma-membrane and the nuclear membrane. In contrast, we observe a signal for STEF following both Triton and digitonin permeabilisation, indicating that some portion of STEF is localised at the cytoplasmic face of the nuclear envelope.

Our biochemical fractionation experiments and immunofluorescence imaging results suggest that STEF is also present in the nucleus, as the reviewer observed. Although these observations are of considerable interest, we believe investigating the function and trafficking of intra-nuclear STEF is beyond the scope of the present study. In addition, our work targeting Rac1 to the perinuclear envelope, as detailed in response to point #10 below, suggests that the perinuclear pool of STEF is important for the regulation of the actin cap.

2) Regarding the fractionation, it is unclear what Woroniuk et al. mean by perinuclear fraction? The protocol described in the method section is a detergent-based fractionation which allows a 1st separation between a fraction (pellet) that contains the membranes (plasma and nuclear membranes) + insoluble cytoskeleton and a fraction (supernatant) that contains the cytoplasmic ("soluble") components. This does not allow specific perinuclear fractionation.

We believe this protocol can generate a fraction enriched in perinuclear proteins, as demonstrated in the original manuscript from which we took the procedure (Shaiken *et al. Scientific Reports*. 2014, 2014. 4:4923). In their manuscript, Shaiken and colleagues performed their protocol in MEFs and

showed high levels of perinuclear proteins including Nesprin 3, Sun2, NUP98/NUP 153 in the 'perinuclear' fraction. But, in light of the reviewer's comments, we have changed the name of this fraction to 'perinuclear-enriched' to indicate that this is not a pure perinuclear fraction.

3) Additionally, the authors used co-IP with nesprin 2G to confirm STEF localization (and STEF interactors), but nesprin 2G is over 700 kD while the figure 1e shows a band at about 300 kD. The signal here is probably unspecific or is due to another (small) isoform of nesprin 2 which may not localize at the nuclear periphery. (Similar comment can be made for Nesprin 1Giant (enaptin) which is over 1,000kD, while the authors show a band at about 250-300 kD in figure 1f) (and the co-IP in supplementary figure 2a). All these make the perinuclear localization of STEF quite questionable.

In this instance, we disagree with the comments of the reviewer for the following reasons:

-The resolution of large proteins is poor, particularly proteins that run above the 250kDa marker on 3-8% TA gels; we have previous evidence of this in our published studies investigating Huwe1, a ~480 kDa protein (Vaughan *et al. Cell Reports*. 2015, 10:88-102). As such large proteins run only short distances above our heaviest molecular weight marker.

-Comparison of our IPs and the data-sheets for 2 different commercially available N2G antibodies show that the 700kDa band is positioned similarly to ours on a western blot. (For more details, please see <http://www.immuquest.com/sub-cellular-markers-c39/nuclear-envelope-c153/nesprin-2-antibody-k20-478-5-p1096>)

-Our siRNA mediated knockdowns of Nesprin-2G in the U2OS cell line using published siRNA sequences (shown below) show a specific loss of this band (and the lower MW bands also observed on the gel), indicating that this band is specific for Nesprin-2G.

-Due to its large size, other published works investigating the function of Nesprin-2G have used the mN2G construct; however, we avoided this approach as over-expression of mN2G downregulates the giant forms of the protein. In addition, mN2G lacks many of the spectrin repeat domains of the giant forms, making it incapable of recapitulating the binding properties of the giant form (i.e. the FHOD1 binding site is not present in mN2G).

We also present further evidence, detailed in response to point #1 above, that strengthen our conclusion that STEF localises to the perinuclear envelope.

4) The authors could also use mutant forms of STEF to identify the signaling domain that allows this peculiar localization (distinct from other related GEFs such as Tiam1, even if Tiam1 can also localize in the nucleus as shown in the supplementary figures).

As prompted by both reviewers, we have expressed a range of HA- tagged deletion mutants of STEF in the U2OS cells, and examined their localisation by immunofluorescence. We found that deletion of the C-terminal PH domain abolished the perinuclear localisation of STEF, while the C-terminal DH-PH domains was sufficient for perinuclear localisation, indicating the importance of the C-terminal PH domain in mediating the localisation of STEF to the nuclear envelope. These data are now included as Fig. 1d-f. The C-terminal PH domain of STEF has only 54% sequence homology with the corresponding domain of Tiam1, which may explain the differential localisation of the two proteins.

We have recently published that Tiam1 does localise in the nucleus (Diamantopoulou *et al. Cancer Cell.* 2017, 31:621-634), but we do not see Tiam1 at the nuclear envelope in either control or STEF-depleted cells (Supplementary Fig. 1f,g).

5) The authors mentioned that STEF is a Rac GEF and the results obtained with the STEF mutant lacking the DH domain suggest that STEF regulates Rac activity at the nuclear periphery. This hypothesis is intriguing and could be tested using Rac FRET sensors to look at perinuclear Rac activity in cells depleted for STEF

We have conducted an experiment using a Raichu-Rac1 FRET biosensor as suggested by the reviewer and observed a significant decrease in perinuclear Rac1 activation in STEF-depleted U2OS cells vs. controls. This is now included in the revised manuscript as Fig. 4a,b.

6) If Rac is activated by STEF how does it regulate perinuclear actin cap? Does it promote NMMIIB activation (myosin phosphorylation)...

The reviewer raised a number of interesting questions relating to the activities downstream of Rac1 which might regulate the perinuclear cap, including NMMIIB (described here), actin polymerisation (point #7), participation in the LINC complex (point #8) and activation of Pak (point #9). We address each of these separately, but in summary, the downstream pathway for which we have the most evidence is phosphorylation of MLC, linking STEF and Rac activity to NMMIIB, and hence to contractility at the nuclear envelope.

In addition to our results that demonstrate a reduction in myosin-generated tension at the nuclear envelope upon depletion of STEF (Fig. 6c,d), we have performed experiments to analyse perinuclear phospho-MLC as a read-out of NMMIIB activity. Through quantification of pMLC-positive perinuclear cables, we showed a significant reduction of perinuclear pMLC in STEF-depleted MEFs, which we have included in the revised manuscript as Fig. 6e,f. These results strengthen our hypothesis that STEF-mediated Rac1 activity regulates the perinuclear actin cap through downstream regulation of NMMIIB activity.

7) or does it stimulate local actin polymerization (via Arp2/3)

We have conducted preliminary experiments to test this hypothesis and saw no effect of Arp2/3 inhibition on the formation of the perinuclear actin cap (data not shown). We also investigated the use of a formin inhibitor (SMIFH2 - CAS 340316-62-3 – Calbiochem, 30uM) on the formation of the perinuclear actin cap and observed a decrease in the number of perinuclear actin cables following treatment with the inhibitor (data included below). However, as it will be difficult to dissect out the contribution of other small GTPases which regulate formins besides Rac1, we have decided not to include this result in the manuscript nor to extend this line of enquiry at this time.

8) ...or alternatively does it participate to the LINC complex assembly?

This is an intriguing hypothesis, and we had conducted preliminary investigations into the relationship between STEF and Nesprin-2G (not included in our manuscript) that would indicate that STEF does not participate in LINC complex assembly. In these experiments, we expressed a mN2G construct in control and STEF depleted MEFs and saw no observable change in the ability of mN2G to localise at the nuclear envelope. Furthermore, we conducted FRAP experiments to determine the effect of STEF depletion on the mobility of Nesprin-2G at the nuclear envelope, investigating if STEF had a role in regulating the anchoring of the LINC complex at the nuclear periphery, or induced global changes to the properties of the nuclear envelope. We observed no change in the t_{1/2} or mobile fraction of mN2G in STEF depleted cells (as shown in the figure below), indicating that STEF does not regulate mN2G mobility. As we observed no change in the localisation or mobility of mN2G in STEF-depleted cells, we did not investigate this hypothesis further and have not included these results in the manuscript as additional experiments would be required to consolidate this point.

C

	Control	STEF KO
Mobile/ Immobile Fraction	0.35/ 0.65	0.37/ 0.62
t _{1/2} (ms)	14679	15496

9) Many pharmacological inhibitors (PAK inhibitors) or plasmid constructs could be used to further analyze the downstream effectors that regulate perinuclear actin cap formation

Our preliminary experiments using a pharmacological inhibitor of PAK (Frax 597) did not recapitulate the phenotype observed in STEF depleted cells (data not shown), therefore we have not investigated this further.

10) Can the authors recapitulate the effect by inactivating Rac (or Rac effector?) Or can they rescue STEF depletion by addressing another Rac GEF (or just the DH-PH domain) and restore Rac activity at the nuclear periphery?

The reviewer addresses a central point of the paper, namely the relationship between the GEF activity of STEF and the local activation of Rac1 at the perinuclear membrane. To address this, we performed experiments to establish the role of perinuclear Rac1 activity in the regulation of the actin cap. We generated a series of Rac1 constructs containing a modified KASH domain derived from Nesprin2, which allows localisation to the nuclear envelope (and somewhat in the endoplasmic reticulum, which is contiguous with the outer nuclear envelope) but does not affect the interaction between Nesprin2 and Sun2 (avoiding the generation of a dominant negative effect, see Fig. 4c in the revised manuscript for a schematic of the construct and Stewart-Hutchinson *et al. Exp Cell Res.* 2008, 314:1892-905). This allowed us to target either WT, active (V12) or dominant negative (N17) Rac1 to the perinuclear membrane. We expressed these constructs in either STEF KO or WT cells and observed their effects on the actin cap.

- Through targeting of constitutively active-Rac1 to the nuclear envelope in STEF KO MEFs, we showed that active Rac1 at the nuclear envelope was sufficient to rescue the disruption of the perinuclear actin cap caused by STEF depletion.
- Expression of WT-Rac1 at the nuclear envelope was not sufficient to rescue the loss of actin cables in STEF KO MEFs, highlighting the requirement for STEF specifically to activate Rac1 at the nuclear periphery and suggesting that no other Rac1 GEF can compensate for its depletion.
- Additionally, targeting of the dominant-negative form of Rac1 to the nuclear envelope caused significant disruption of the actin cap in WT MEFs, further proof of Rac's role in regulating the actin cap.

We feel that these experiments together confirm that the disruption of the actin cap upon depletion of STEF is indeed a Rac1-dependent phenotype, and furthermore highlight the importance of STEF-mediated localised Rac1 activity at the nuclear envelope in actin cap regulation. We have now included these as Fig. 4c-g in our revised manuscript.

11) ...the decreased nuclear stiffness could also be the consequence of an altered lamina...did the authors look at lamin A/C and lamin B expression and localization in cells depleted for STEF?"

To address this point, we investigated the expression levels of Lamin A/C and Lamin B proteins in control and STEF-depleted MEFs and saw no noticeable change. We present this data in the revised manuscript as Supplementary Fig. 6a,b. We also include in Supplementary Fig. 6c representative immunofluorescence images of parental and STEF-depleted U2OS cells stained for STEF, Lamin A/C and Hoechst which show no apparent change in the localisation or intensity of Lamin A/C upon down-regulation of STEF.

12) The authors analyzed TAZ nuclear expression (Sup figure 5) and conclude that STEF regulates TAZ activity, although usually one measures the nuclear/cytoplasmic ratio of TAZ to investigate its

activity. Without doing any new experiments, the authors should re-analyze the images and see if the ratio is altered.

We attempted nuclear/cytoplasmic biochemical fractionation experiments to corroborate the immunofluorescence images present in our original manuscript, but saw no statistically significant effect of STEF depletion on the nuclear/cytoplasmic ratio of TAZ. For these reasons we have removed the data relating to the nucleo-cytoplasmic shuttling of TAZ from the manuscript. However, determining the nuclear/cytoplasmic ratio of TAZ can be complicated by TAZ degradation in the cytoplasm, so that the nuclear/cytoplasmic ratio is not always indicative of TAZ activity. Moreover, not all control of TAZ activity is through regulating nucleo-cytoplasmic shuttling (for example see our recent publication Diamantopoulou *et al. Cancer Cell.* 2017, 31:621-634). Nonetheless, to expand our investigation of the link between STEF and TAZ, we performed qPCR analysis of the expression levels of the TAZ-regulated gene *Ctgf* in liver tissue isolated from WT and STEF KO mice. We observed a significant reduction in *Ctgf* expression levels in the STEF KO samples (now included in the manuscript as Fig. 6h), consistent with the results of similar experiments performed in WT and KO MEFs (Fig. 6g and Supplementary Fig. 6c), indicating a clear link between STEF depletion and YAP/TAZ activity.

13) What do the authors mean by “nuclear morphology” in the title? Nuclear height is affected by STEF depletion, but morphology can relate to so many aspects and having it in the title could be misleading.

We agree that “nuclear morphology” can encompass many properties of the nucleus, whereas we are describing changes specifically in nuclear height and orientation. We have therefore altered the title of the revised manuscript to address this point and to reflect our additional data confirming the importance of perinuclear Rac1 activity in the regulation of the actin cap.

14) Since a confocal microscope was used for imaging actin in figure 3 b and e, the authors should show a representative z projection view to allow better characterization of the perinuclear actin network (fig3b 3e)

These images were obtained using a widefield (Deltavision) microscope; we apologise for this mistake in the text of the original manuscript. We have now included single z-stack images of the basal actin fibres from these images, and quantified basal actin (Supplementary Fig. 3d,e). The Rac1 targeting rescue experiments were analysed using a Zeiss Airyscan confocal microscope, and as such we now include single z-plane images of apical and basal actin from control and STEF knockout cells (Fig. 4d and Supplementary Fig. 4e). These data together show more clearly that STEF depletion does not have a global effect on actin organisation; we see a specific disruption of apical actin cables with no significant effect on basal actin structures.

15) Ref 23- this reference may not be adequately used. In this paper published by Hahn and colleagues, we can observe perinuclear Rac-GFP, but very low perinuclear Rac activity (using FRET sensors) is reported in this paper.

To clarify this comment, we include below Fig. 4 from the Kraynov *et al.* manuscript which shows both the FRET signal and GFP-Rac signal in motile Swiss 3T3 cells. As noted in the figure legend, the ‘highest concentration of activated Rac1 was seen in the juxtannuclear area’ and regions of perinuclear signal can be visualised on the images included. Furthermore, during our recent collaboration with Klaus Hahn’s laboratory we have confirmed our interpretation of their manuscript and conducted further FRET experiments examining the activation levels of perinuclear Rac1 in control and STEF-depleted cells as mentioned above. We observed a significant decrease in

perinuclear Rac1 activation in STEF-depleted U2OS cells vs. controls and this experiment is now included in the revised manuscript as Fig. 4a,b.

Fig. 4. Rac nucleotide state in motile cells. **(A)** Two examples of Rac activation and localization in motile Swiss 3T3 fibroblasts (bar = 24 μ m). Cells were induced to move by scraping a wound in a cell monolayer (20). The highest concentration of activated Rac1 was seen in the juxtanuclear region, and a gradient of Rac activation was also observed, highest near the leading edge and tapering off toward the nucleus. Color scale for the intensity of FRET or GFP fluorescence is the same as Fig. 2C. FRET intensities are 0 to 18 (top image) and 0 to 32 (bottom image). In the GFP images, intensities range from 98 to 700 (top image) and 100 to 1100 (bottom image). **(B)** Example of a cell in the monolayer, away from the wound. In such cells, FRET was either not detected or found around the cell edge. (GFP intensities = 0 to 1023, FRET intensities = 0 to 10).

16) The co IP STEF/ NMMIIB is hard to distinguish since the two proteins have similar molecular weight, additional view of the uncropped gels as supplementary could be useful.

Please see in Supplementary Fig. 7 the uncropped blots of the co IP for STEF and NMMIIB included as Fig. 2f of the manuscript.

REVIEWERS' COMMENTS:

Reviewer #1 (Remarks to the Author):

The authors have provided a very detailed rebuttal, and have also made many significant changes to the manuscript. I am convinced that STEF does have a different localisation to Tiam1, and as such it may well be playing a role in the activation of Rac1 to control the structure of the actin cap.

A confusing issue in the literature is that STEF/ Tiam2 was annotated as Tiam1 in early papers (cf. EMBO J. 2003 Aug 15;22(16):4190-201. So I am pleased that the authors address this issue in their title.

Reviewer #2 (Remarks to the Author):

The authors have addressed most of the comments and have substantially improved the manuscript. However, a few details remain unclear and should be specified and/or modified in the manuscript. I realize that these details can be perceived as trivial or insignificant, but the cell biology of the nucleus is a rapidly growing field and it would be very valuable to use an adequate nomenclature, especially for publication in Nature Communications.

-Nesprin 2 molecular weight - It is certainly hard to guarantee the molecular weight of large proteins in western blot, but it is important to know which isoform is observed by the authors here. Since the highest molecular weight marker is 250 kD, can the authors add a blot of a protein with similar molecular weight (on the same membrane)? (Why the authors have removed nesprin 1G? Which was in the previous figure 1 and is still in the antibody list).

-Perinuclear fraction – The issue is not the purity of this preparation but its nomenclature. The staining performed with digitonin permeabilization is very convincing and show that STEF localizes at the nuclear periphery (perinuclear staining). Although the “perinuclear” fraction (figure 1g) contains proteins from the INM (Shaiken et al., ref cited by the authors) and from the lamina (lamin B), which do not belong to what one would call “perinuclear” fraction. The authors should consider defining clearly what they call perinuclear or changing this name to a more appropriate one. (It is also surprising to use “perinuclear envelope”, instead of nuclear envelope)

-Regarding the pool of perinuclear active Rac, the data included in this revised version are very convincing (figure 4a). However in the paper from Kraynov et al., there is no “perinuclear” active

Rac. We can see active Rac in the posterior 1/3 area between the cell front and the nucleus (what the authors called juxta nuclear, which is definitely not perinuclear).

REVIEWERS' COMMENTS:

Reviewer #1 (Remarks to the Author):

The authors have provided a very detailed rebuttal, and have also made many significant changes to the manuscript. I am convinced that STEF does have a different localisation to Tiam1, and as such it may well be playing a role in the activation of Rac1 to control the structure of the actin cap.

We are pleased that the reviewer is now convinced of the differential localisation of STEF and Tiam1, and of our hypothesis that STEF plays a role in regulating Rac1 activation to regulate the actin cap.

A confusing issue in the literature is that STEF/ Tiam2 was annotated as Tiam1 in early papers (cf. EMBO J. 2003 Aug 15;22(16):4190-201. So I am pleased that the authors address this issue in their title.

We are pleased to have helped clarify this confusion.

Reviewer #2 (Remarks to the Author):

The authors have addressed most of the comments and have substantially improved the manuscript. However, a few details remain unclear and should be specified and/or modified in the manuscript. I realize that these details can be perceived as trivial or insignificant, but the cell biology of the nucleus is a rapidly growing field and it would be very valuable to use an adequate nomenclature, especially for publication in Nature Communications.

We were pleased to read that the reviewer is happy that we have addressed most of their comments with our revisions to the manuscript. We appreciate that in a field which is rapidly growing it is important to be as precise as possible with definitions and nomenclature.

-Nesprin 2 molecular weight - It is certainly hard to guarantee the molecular weight of large proteins in western blot, but it is important to know which isoform is observed by the authors here. Since the highest molecular weight marker is 250 kD, can the authors add a blot of a protein with similar molecular weight (on the same membrane)?

We have added images from a single blot of U2OS lysates probed for Nesprin2 (796kDa) and for Huwe1 (482 kDa), a large protein we have studied in previous publications and for which we have validated the antibody. As can be seen in Supplementary Figure 2b, the band of Nesprin2 runs higher than that of Huwe1, and both are substantially above the 250 kD marker.

To further validate the antibody for Nesprin 2 we have added a western blot showing knockdown of Nesprin2 protein by two siRNAs to this supplementary figure (Supplementary Figure 2a). We have referenced both of these in the text.

(Why the authors have removed nesprin 1G? Which was in the previous figure 1 and is still in the antibody list).

This figure has been moved from the main Figure 1 into Supplementary Figure 1 for clarity.

-Perinuclear fraction – The issue is not the purity of this preparation but its nomenclature. The staining performed with digitonin permeabilization is very convincing and show that STEF localizes at the nuclear periphery (perinuclear staining). Although the “perinuclear” fraction (figure 1g) contains proteins from the INM (Shaiken et al., ref cited by the authors) and from the lamina (lamin B), which do not belong to what one would call “perinuclear” fraction. The authors should consider defining clearly what they call perinuclear or changing this name to a more appropriate one.

We have changed the title of this fraction to “perinuclear enriched” to reflect the reviewers concerns on this issue.

(It is also surprising to use “perinuclear envelope”, instead of nuclear envelope)

We searched the manuscript, but could not find the term “perinuclear envelope”.

-Regarding the pool of perinuclear active Rac, the data included in this revised version are very convincing (figure 4a). However in the paper from Kraynov et al., there is no “perinuclear” active Rac. We can see active Rac in the posterior 1/3 area between the cell front and the nucleus (what the authors called juxta nuclear, which is definitely not perinuclear).

We agree that the juxta nuclear localisation of Rac1 in the Kraynov et al paper is different from the localisation we have seen (Figure 4a). When we refer to the Kraynov et al paper in our manuscript active Rac1 is described as juxta nuclear.